



# Ages and transit times as important diagnostics of model performance for predicting carbon dynamics in terrestrial vegetation models

Verónika Ceballos-Núñez[1], Andrew D. Richardson[2, 3, 4], and Carlos A. Sierra[1]

[1]Max Planck Institute for Biogeochemistry, Hans-Knöll-Str. 10, 07745 Jena, Germany
[2]Department of Organismic and Evolutionary Biology, Harvard University, 02138 Cambridge, MA, USA
[3]School of Informatics, Computing and Cyber Systems, Northern Arizona University, Flagstaff, AZ 86011, USA.
[4]Center for Ecosystem Science and Society, Northern Arizona University, Flagstaff, AZ 86011, USA.

*Correspondence to:* Carlos A. Sierra (csierra@bgc-jena.mpg.de)

**Abstract.** The global carbon cycle is strongly controlled by the source/sink strength of vegetation as well as the capacity of terrestrial ecosystems to retain this carbon. These dynamics, as well as processes such as the mixing of old and newly fixed carbon, have been studied using ecosystem models, but different assumptions regarding the carbon allocation strategies and other model structures may result in highly divergent model predictions. We modeled three systems of vegetation compartments and assessed their performance by calculating the age of the carbon in vegetation system and within each compartment, and the overall transit time of C in the system. We used these diagnostics to assess the influence of three different carbon allocation schemes on the rates of C cycling in vegetation. First, we used published measurements of ecosystem C compartments from the Harvard Forest Environmental Measurement Site to find the best set of parameters for the different model structures. Second, we calculated C stocks, release fluxes, radiocarbon values based on the bomb spike, ages, and transit times. We found a good fit of the three model structures to the available data, but the time series of C in foliage and wood need to be complemented with other ecosystem compartments in order to reduce the high parameter collinearity that we observed, and reduce model equifinality. Differences in model structures had a small impact on predicting C stocks in ecosystem compartments, but overall they resulted in very different predictions of age and transit time distributions. In particular, the inclusion of two storage compartments resulted in the prediction of a system mean age that was 10-20 years older than in the models with one or no storage compartments. The age of carbon in the wood compartment of this model was also distributed towards older ages, whereas fast cycling compartments had an age distribution that did not exceed 5 years. As expected, models with C distributed towards older ages also had longer transit times. These results suggest that ages and transit times, which can be indirectly measured using isotope tracers, serve as important diagnostics of model structure and could largely help to reduce uncertainties in model predictions. Furthermore, by considering age and transit times of C in vegetation compartments as distributions, not only their mean values, we obtain additional insights on the temporal dynamics of carbon use, storage, and allocation to plant parts, which not only depends on the rate at which this C is transferred in and out of the compartments, but also on the stochastic nature of the process itself.



# 1   Introduction

The global carbon cycle is strongly controlled by the source/sink strength of terrestrial ecosystems. Vegetation in particular, is one of the major controls of global C sources and sinks with respect to the atmosphere (Canadell et al., 2007); it has the capacity to be either a strong C sink or a source, depending on the amount of C fixed by the canopy and the time that C takes

to transit through its components back to the atmosphere (Luo et al., 2003). Strong sinks therefore, not only fix carbon at a fast rate, but have also the capacity to store this carbon for long periods of time (Körner, 2017).

The C storage capacity of an ecosystem is determined by the collective behavior of vegetation compartments such as foliage, wood, and roots, which may also act as C sources and sinks among each other (Xia et al., 2013; Luo et al., 2017). The capacity of a vegetation compartment to oscillate between C source and sink has important implications for ecosystems to respond to

perturbations and environmental change, i.e their resilience. Carbon fixed during photosynthesis is transported from the leaves (sources) to other parts of the plant (sinks). One of these sinks is the non-structural carbon (NSC) (Hartmann and Trumbore, 2016; Trumbore et al., 2015; Martínez-Vilalta et al., 2016), which may turn into a C source during critical events, such as the start of the growing season (after periods of limited photosynthesis) (Richardson et al., 2013), and the recovery from disturbances such as drought (Hartmann et al., 2013), cold temperatures (Hoch and Körner, 2003), pollution (Grulke et al.,

2001), or nutrient stress (Ericsson et al., 1996). Despite the importance of the source-sink capacity of NSC reserves, many questions remain unsolved. For example, are NSCs completely depleted when needed, and replenished afterwards? Is the C that has remained stored for many years still available for the plant? (Richardson et al., 2013).

It is indeed possible that the carbon stored in vegetation compartments, including NSCs, have been fixed at different times, resulting in a mix of ages (Muhr et al., 2016). Studies across wood rings in temperate forest trees revealed that the mean age of

NSCs in stemwood can be up to several decades old (Richardson et al., 2013; Trumbore et al., 2015). Trumbore et al. (2015) explained these old ages with a simple model consisting of one NSC compartment with inward mixing of younger and older C. Alternatively, Richardson et al. (2013) proposed a model with two separate storage compartments (NSC)-with old and young C, respectively- that exchange material among each other. It is therefore uncertain how this mixing of NSCs of different ages occurs: In the form of one single compartment in which all ages are mixed, or in different compartments with separate ages?

Previous studies have focused mostly on determining ages of NSCs using radiocarbon-derived mean residence times, but this approach has limitations. One limitation is the ambiguity in the term "mean residence time", which has been defined in different ways across studies; in some cases it implies the mean age of C in an ecosystem or ecosystem compartment and in other cases it implies the time it takes C molecules to leave the system of compartments (Sierra et al., 2016). Another limitation is the use of mean values instead of complete frequency or density distributions to assess the spread of C ages in vegetation

compartments.

Certainly the study of C age distribution in vegetation requires challenging empirical methods, but can also be approached using ecosystem C cycle models. However, not all of the models perform equally well because the assumptions behind their structures may result in highly divergent predictions (Lacointe, 2000; Friedlingstein et al., 2006; Friend et al., 2014; Schiestl-Aalto et al., 2015). The performance of such models has been diagnosed by comparing their predicted C storage capacity and




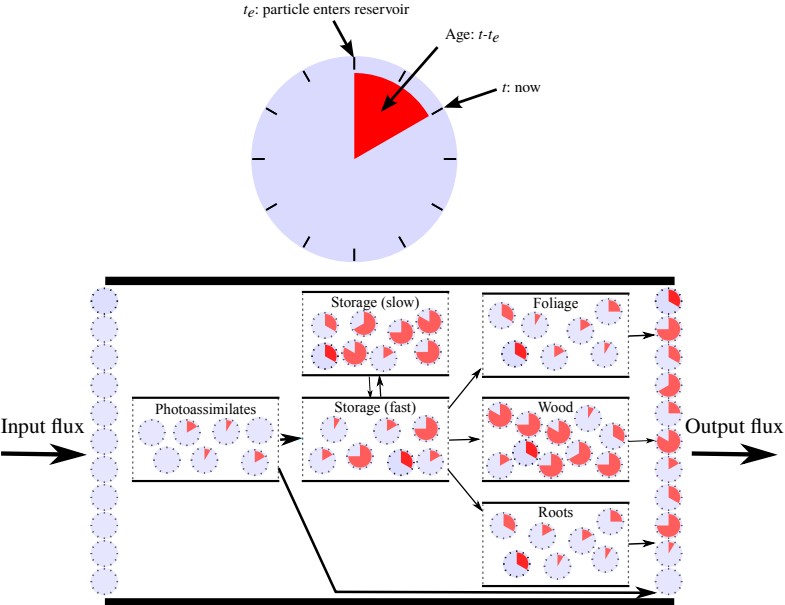

**Figure 1.** Graphical representation of the concepts of age and transit time distributions in a vegetation model. Carbon particles are represented here as clocks that measure the time they have been in the system. **System age** can be defined as the age of all particles in the system at a given time, while **transit time** as the age of particles in the output flux. Adapted from (Sierra et al., 2016)

residence times (Friend et al., 2014; Yizhao et al., 2015), but if the above mentioned ambiguities are resolved, ages of carbon in vegetation and in respiration fluxes can serve as excellent additional diagnostics of ecosystem models, and can give important insights about carbon metabolism in vegetation under stress conditions, such as in the case of drought stress.

## 1.1 Definitions: ages and transit times

$CO_2$ molecules are fixed continuously by photosynthesis during the growing season; so, C particles enter the vegetation (from here on called "system") at different times during the year. After fixation, the photosynthetic products transit through the vegetation compartments until they eventually leave the system, either as $CO_2$ back to the atmosphere or as litter and exudates to the soil. This means, that at a given time ($t$) each particle of the system has a different *age* , which is the time that it has remained in the system since its fixation from the atmosphere. The time that each particle spends transiting through the system, from arrival until exit, is called *transit time* (Bolin and Rodhe, 1973).

Although C particles follow random paths within a system, the speed at which C flows into or out of the system and from one compartment to another is given by certain rates (or transfer coefficients). This means that the age of carbon in a system is determined by stochastic and deterministic processes, which can be illustrated in Figure 1; given two organic molecules in a compartment, one that has remained in the system for longer time than the other, they both have the same chance to either leave the system, or move on to another compartment. Thus, if by chance older molecules remain for longer times, then the





system's age gets older. But the pace at which these molecules are transiting is moderated by the cycling rates. This is why slow cycling compartments have older C.

Lets consider a system of well-mixed C, distributed in multiple compartments. We can describe it with the following system of ordinary differential equations 1:

$$\dot{\boldsymbol{x}}(t) = \mathbf{B} \times \boldsymbol{x}(\boldsymbol{t}) + \boldsymbol{\beta} \cdot u,$$
$$\boldsymbol{x}(0) = \boldsymbol{x}_0,$$
(1)

where $\dot{\boldsymbol{x}}(t)$ is how much the quantity of carbon in vegetation compartment $x$ changes with respect to time, $\mathbf{B}$ is a matrix of carbon transfer coefficients between the plant compartments, $\boldsymbol{x}(t)$ is a vector of states for vegetation (state variables), $\boldsymbol{\beta}$ is a

vector containing the partitioning coefficients of photosynthetic input, and $u$ is a scalar that represents that input. This linear system does not include environmental variables, or any other variables that depend on time, thus, it is an *autonomous linear system* with multiple interconnected compartments.

Given that each particle in the system has its own age and transit time, the age and transit time of the whole system can be considered as random variables. Additionally, the age and transit time of particles in a system's compartment is exponentially

distributed. Then, the age and transit time distributions of the system would be the sum of those exponential distributions, i.e. a phase-type distribution (PT) (Metzler and Sierra, 2017).

The calculation of how many C particles have a certain age, or *the age density distribution of a system* ($f_A(y)$), is determined by the probability of entering the system through a given compartment and the rates at which C is transferred from one compartment to another until it leaves the system. Consistent with the symbols from the previous equation

$$f_A(y) = \boldsymbol{z}^T \cdot e^{y \cdot \mathbf{B}} \cdot \frac{\boldsymbol{x}^*}{||\boldsymbol{x}^*||}.$$
(2)

$f_A(y))$ is a function of (i) how fast the carbon is leaving the system: the row vector of release rates, which is the column-wise sum of the elements of the matrix $\mathbf{B}$ ($\boldsymbol{z}^T = -\mathbf{1}^T \mathbf{B}$), (ii) the transition probability matrix ($e^{y\mathbf{B}}$), and (iii) the relative amount of C stock at steady state with respect to the total ($\frac{\boldsymbol{x}^*}{||\boldsymbol{x}^*||}$). Notice that we use here the symbol $|| \cdot ||$ to represent the vector norm, which is the sum of all entries of the vector.

The mean age is given by the expected value ($\mathbb{E}[A]$)

$$\mathbb{E}[A] = \frac{||\mathbf{B}^{-1} \times \boldsymbol{x}^*||}{||\boldsymbol{x}^*||}.$$
(3)

Likewise, the transit time density distribution ($f_{FTT}(t)$) is also a function of $\boldsymbol{z}^T$ and the transition probability matrix ($e^{t\mathbf{B}}$), as well as the vector of input distributions ($\boldsymbol{\beta}$).

$$f_{FTT}(t) = \boldsymbol{z}^T \cdot e^{t \cdot \mathbf{B}} \cdot \boldsymbol{\beta}.$$
(4)





The mean transit time is defined as ($\mathbb{E}[FTT]$):

$$\mathbb{E}[FTT] = ||\mathbf{B}^{-1} \times \boldsymbol{\beta}|| = \frac{||\boldsymbol{x}^*||}{||\boldsymbol{u}||} \tag{5}$$

In this case, the definition of mean transit time coincides with the commonly used *stock over flux* approach (turnover time), but note that the definitions presented here can only be applied to autonomous systems at steady state. There are other formulas for the age and transit time distributions of non-autonomous models (e.g., Rasmussen et al., 2016).

From these equations it is evident that age and transit time calculations mainly depend on the schemes of C partitioning ($\boldsymbol{\beta}$) and cycling (**B**) within a vegetation model. Therefore, if we want to understand processes such as the mixing of old and newly fixed NSC using ecosystem models, it is critical to model proper carbon allocation (CA) strategies. Unfortunately, it is still uncertain what assumptions and simplifications should be done: How many carbon compartments are necessary to describe carbon cycling in vegetation? How are these compartments interconnected and how fast are they transferring C among each

other? These are important questions that need to be addressed to improve our understanding of vegetation dynamics, and predict consequences of environmental change on vegetation.

In this contribution, we address the question: how different C allocation schemes affect the ages and transit times of carbon in vegetation models? In particular, we are interested in understanding whether different carbon allocation strategies would lead to different patterns of mixing of ages for the NSC compartment. For this work, we implemented 3 carbon allocation schemes

based on (Richardson et al., 2013); the models have either no storage, 1 storage compartment, or 2 storage compartments (fast and slow C cycling). First, we used published measurements of ecosystem C compartments from the Harvard Forest Environmental Measurement Site to find the best set of parameters for the different model structures. And second, we diagnosed the performance of these models using as metrics 1) C release fluxes (respiration and other carbon losses such as litterfall), 2) the dynamics of radiocarbon (based on the bomb spike) for individual compartments, 3) the transit time distribution of the

system, and 4) the age distribution of C in the system and in each compartment.

## 2  Methods

We implemented three models whose carbon allocation strategies varied depending on the number of storage compartments (0, 1, or 2) (Figure 2), following the hypotheses proposed by Richardson et al. (2013). The core structure of the models is a constant photosynthetic input -gross primary production (GPP): $u$-of 1400 $gCm^{-2}year^{-1}$ (Urbanski et al., 2007), which enters

the system through the Photoassimilates compartment. Part of the C in this compartment is released back to the atmosphere at each time step, in a flux proportional to the size of Photoassimilates and the constant rate $R_a$; the environmental variables (which operate in an hourly-daily time basis) were not not considered because we ran the models at an annual time scale; i.e., without phenology. In the model without storage compartment, the C stored in the Photoassimilates is partitioned into Structural foliage (from here on: Str. Foliage), Wood (including branches and coarse roots) and Fine roots (from here on: Roots), with the constant rates $A_f$, $A_w$, and $A_r$; part of the C stored in these three compartments also leaves the system with





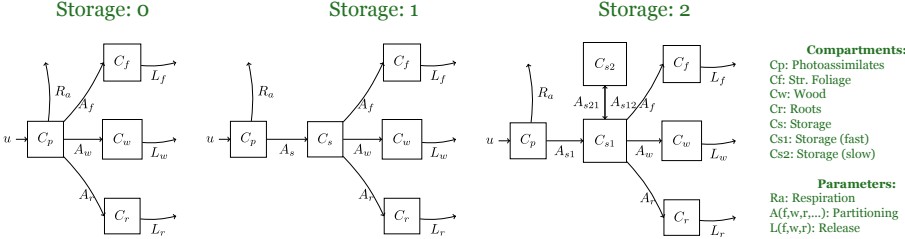

**Figure 2.** Three carbon allocation strategies in vegetation models. These strategies differ in the number of storage compartments, Storage: 0, Storage: 1, and Storage: 2. Adapted from (Richardson et al., 2013). The parameters are the rates at which the carbon cycles into and out of the compartments; thus, the fluxes are proportional to the C stocks and the rates.

constant rates $L_f$, $L_w$, and $L_r$, which comprise all the carbon released through respiration and other losses (e.g. litterfall). For the models with storage, the C is transferred from the Photoassimilates to the fast cycling storage, from which it is then partitioned to the rest of the compartments. In addition to the fast cycling storage, the model with 2 storage compartments also has a slow cycling compartment. All C transfers between these compartments depend on constant rates. These models are autonomous (no time dependencies) linear systems with multiple interconnected compartments, and therefore they meet the

requirements of equation 1. They were implemented in R using the R package `SoilR` (Sierra et al., 2012).

The initial parameter values were obtained from the literature, from a deciduous-evergreen model with similar carbon allocation schemes (Fox et al., 2009). However, in order to find the best set of parameters for these models, we performed a parameter estimation procedure using published measurements of two ecosystem C compartments from the Harvard Forest Environmental Measurement Site, AmeriFlux US-Ha1 Harvard Forest EMS Tower (HFR1), US-Ha1: Harvard Forest EMS Tower (HFR1),

and the raw LAI . Harvard Forest is a regenerating temperate forest located in Petersham, Massachusetts (42.54°N, 72.18°W, 340 m asl); among the tree species that are found in this 65- to 85-year-old mixed deciduous forest are: red oak, red maple, white and red pine, yellow and white birch, beech, ash, sugar maple and hemlock (Wofsy et al., 1993). We performed a two-step optimization procedure using first a classical optimization method and plugging in the results into a Bayesian optimization using the *R* package `FME` (Soetaert and Petzoldt, 2010). Taking into account that data uncertainties have a direct influence on

the fit of the outcome of the parameter estimation (Richardson et al., 2010), we accounted for the uncertainty in the data using the standard deviation of the measurements in the cost functions.

Since the lack of constraints for the other compartments resulted in unrealistic C stocks, we used the above mentioned data to calculate the C stocks in Roots and Storage and further constrain the parameter estimation. For this purpose, we relied on the assumption that shoot:root ratio = 1:5, and the NSC calculations from Wood (Richardson et al., 2010, p. 5).

As means to evaluate whether the parameters could be estimated from the given data sets, i.e. parameter identifiability, we performed a local sensitivity analysis and estimated the collinearity of the parameter sets with the package `FME`. The obtained collinearity index $\gamma$ expresses the degree at which pairs of parameters are linearly related. Values of $\gamma > 20$ indicate high collinearity among parameters and poor identifiability of the model given the available data (Soetaert and Petzoldt, 2010).




5 Since some of the parameters were highly correlated, we ran all model simulations using the posterior parameter set with the highest frequency.

With the obtained parameter sets, we calculated C stocks, release fluxes, radiocarbon values based on the bomb spike, ages, and transit times, using functions implemented in `SoilR`. The functions that calculate age and transit time distributions are based on the formulas proposed by Metzler and Sierra (2017).

## 2.1 Uncertainty analysis

In order to explore model predictions that could result from different parameter sets that were possible and likely, we extracted a random sample of 1000 posterior parameter sets from the Bayesian optimization that used Markov chain Monte Carlo. We ran the models with the unique sets, and calculated the weighted mean and standard deviation of the C stocks, the released C from each compartment, and the system's mean age and transit time. The weights corresponded to the number of repetitions of 15 each unique parameter set in the sample.

In summary, we implemented three models with carbon allocation strategies that varied depending on the number of storage compartments: *Storage: 0*, *Storage: 1*, and *Storage: 2*. We estimated the parameter values using available published data, and then we calculated the local sensitivity analysis and collinearities of the parameters. Finally we assessed the influence of the carbon allocation strategies on ecosystem level C cycling using different metrics.

20 *Code and data availability.* All of the simulations and figures for this work can be reproduced using the code and data provided in the supplementary material.

## 3 Results

### 3.1 Parameter estimation, local sensitivity analysis, and collinearities

Model prediction using the best parameter sets estimated from both optimization procedures (Table 1) provided a good fit to the 25 available data. The simulations of C stocks in Wood were in accordance with the aboveground biomass inventory data (Figure 3). Predictions of C stocks in Foliage (Photoassimilates + Structural) were also as expected from the data (4).

Despite the good fit, some of the parameters were strongly correlated among each other. For the three model structures and empirical data, the number of parameters that can be simultaneously estimated with a collinearity index $< 20$ was 3 for the model without storage, and 5 for the other models. The correlations can be seen in the pairwise plots of sensitivity functions (Figures A1-A3). Table 2 summarizes the number of parameter correlations that we observed in the lower-diagonal of the pairwise plots of sensitivity functions.



**Table 1.** Parameter values obtained from the optimization procedures [$\text{year}^{-1}$].

| Model | Parameter | Final | Best1 | Best2 | Median | $q_{25}$ | $q_{75}$ |
|-------|-----------|-------|-------|-------|--------|----------|----------|
| *Storage: 0* | Ra | 0.28 | 0.7 | 0.7 | 0.6 | 0.52 | 0.66 |
|  | Af | 0.48 | 0.5 | 0.5 | 0.4 | 0.33 | 0.46 |
|  | Ar | 0.44 | 0.5 | 0.5 | 0.39 | 0.31 | 0.45 |
|  | Aw | 0.49 | 0.5 | 0.5 | 0.41 | 0.33 | 0.46 |
|  | Lf | 23.32 | 35.35 | 35.35 | 19.82 | 12.65 | 27.72 |
|  | Lr | 3.03 | 2.71 | 2.71 | 2.67 | 2.23 | 3.09 |
|  | Lw | 0.04 | 0.02 | 0.02 | 0.02 | 0.02 | 0.03 |
|  |  |  |  |  |  |  |  |
| *Storage: 1* | Ra | 0.67 | 0.5 | 0.33 | 0.45 | 0.34 | 0.56 |
|  | Af | 0.27 | 0.19 | 0.4 | 0.36 | 0.24 | 0.44 |
|  | Ar | 0.12 | 0.4 | 0.32 | 0.25 | 0.18 | 0.32 |
|  | Aw | 0.25 | 0.38 | 0.24 | 0.38 | 0.31 | 0.44 |
|  | Lf | 10.43 | 18.25 | 28.93 | 16.12 | 8.87 | 24.97 |
|  | Lr | 1.63 | 3.27 | 3.39 | 2.73 | 2.03 | 3.2 |
|  | Lw | 0.03 | 0.03 | 0.02 | 0.04 | 0.03 | 0.05 |
|  | As | 1.07 | 2.02 | 2.23 | 2.29 | 1.92 | 2.89 |
|  |  |  |  |  |  |  |  |
| *Storage: 2* | Ra | 0.65 | 0.64 | 0.52 | 0.58 | 0.5 | 0.65 |
|  | Af | 0.36 | 0.3 | 0.5 | 0.43 | 0.38 | 0.47 |
|  | Ar | 0.26 | 0.47 | 0.48 | 0.4 | 0.33 | 0.45 |
|  | Aw | 0.24 | 0.49 | 0.47 | 0.34 | 0.27 | 0.4 |
|  | Lf | 12.88 | 35.63 | 27.44 | 16.48 | 10.28 | 26.96 |
|  | Lr | 2.47 | 2.19 | 3.15 | 2.96 | 2.52 | 3.3 |
|  | Lw | 0.02 | 0.01 | 0.03 | 0.02 | 0.02 | 0.03 |
|  | As | 1.92 | 1.99 | 1.96 | 1.75 | 1.5 | 2.13 |
|  | As12 | 0 | 0.15 | 0.02 | 0.01 | 0.01 | 0.02 |
|  | As21 | 0.02 | 0.01 | 0.03 | 0.04 | 0.03 | 0.07 |

Final: Parameter sets used for all of the simulations, unless otherwise noted.

Best1: Parameter set obtained from the classical optimization procedure.

Best2 and the remaining columns were the result of the Bayesian optimization.

## 3.2 Influence of carbon allocation strategies on ecosystem level C cycling

To assess the impact of different carbon allocation strategies on the ecosystem C cycling, we used the following metrics: 1) C release fluxes, 2) dynamics of radiocarbon for individual compartments, 3) transit time distribution of C through the system,





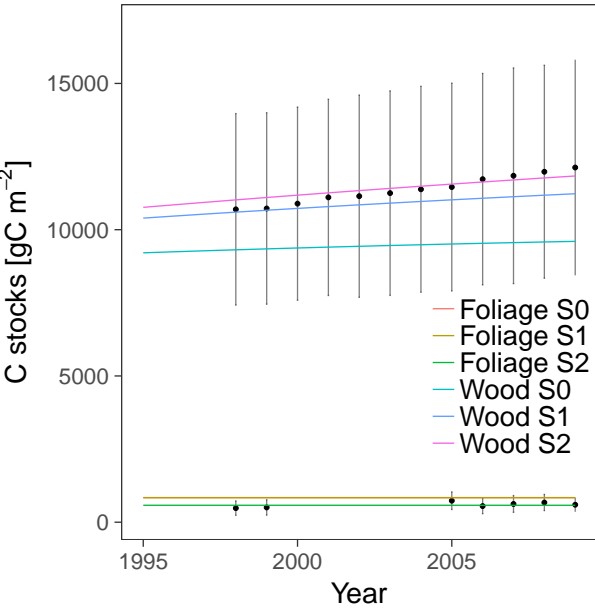

**Figure 3.** Carbon stocks estimated for each model, comparing the observed data and model predictions for the years with available data. These simulations were obtained using the best parameter set.

**Table 2.** Number of positive and negative correlations between parameters

| Model | Positive correlations | Negative correlations | Possible combinations |
|---|---|---|---|
| *Storage: 0* | 5 | 12 | 21 |
| *Storage: 1* | 4 | 10 | 28 |
| *Storage: 2* | 16 | 16 | 45 |

5    and 4) age distribution of C in the system and in each compartment. The calculations required for these metrics were performed using the best values of the parameter sets that resulted from the Bayesian optimization, unless otherwise noted.

### 3.2.1  Fluxes of C released from the compartments

The tree models predicted different mean fluxes of C released from each compartment at steady state (Figure 5). The only exception was the Roots, which had large uncertainties and overlaps among the flux distributions of the three models. This means that certain combinations of model structures with parameter sets result in similar predictions of Root C release fluxes. However, for the compartments related to the foliage the differences were stronger. Thus, regardless of the parameter sets, the differences in model structure lead to the prediction of different C release fluxes at steady state.





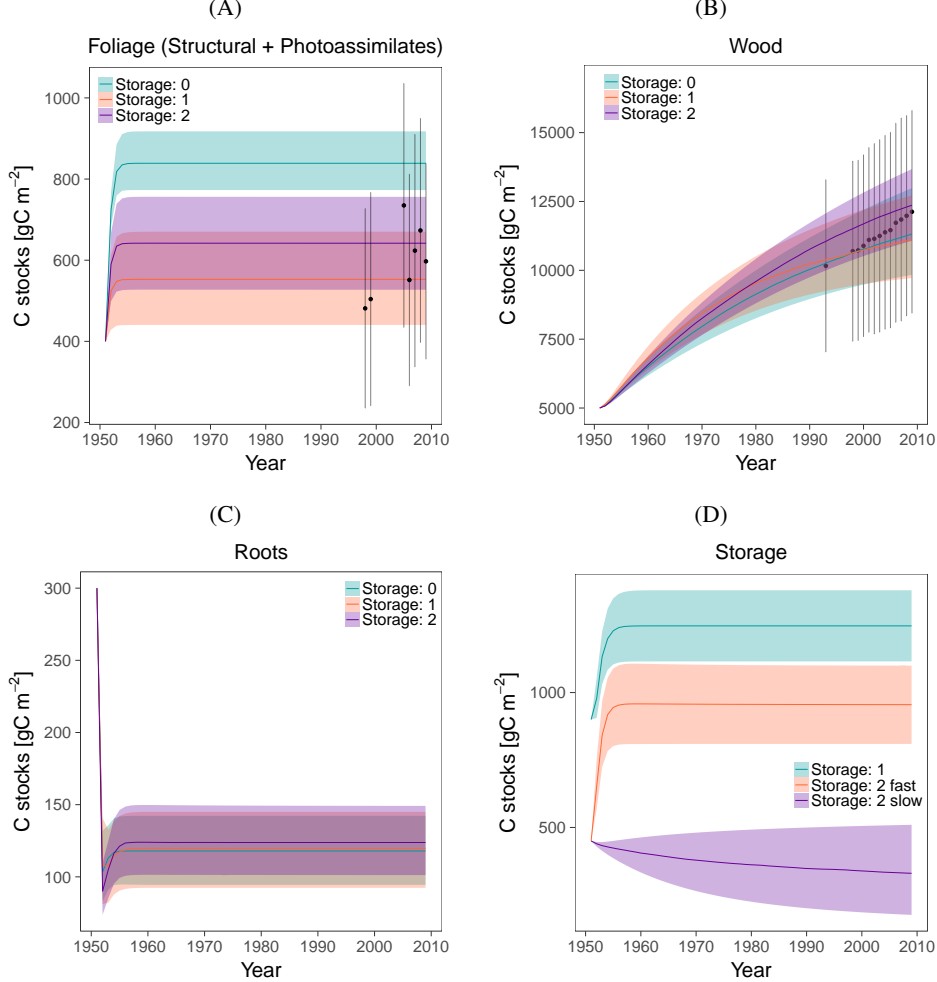

**Figure 4.** Carbon stocks estimated for each compartment and their uncertainties. Carbon in the (A) Foliage (Photoassimilates + Structural), (B) Wood, (C) Roots, and (D) Storage compartments.

### 3.2.2 Radiocarbon content in each compartment

5 The simulated radiocarbon content of fast cycling compartments (e.g. Photoassimilates, Str. Foliage, Storage (fast), and Roots) had a stronger resemblance to the atmospheric $\Delta^{14}C$ values than the slower cycling compartments (Figure 6). However, for the Str. Foliage of the models with storage there was a time lag of about 3 years with respect to the peak that corresponds to the 'bomb-spike'. This short time lag might be the result of that lack of phenology in the models. Furthermore, the accumulation of radiocarbon in slow cycling compartments such as Wood and the Storage (slow), was characterized by a slow incorporation of radiocarbon that resulted in large $\Delta^{14}C$ values of the last part of the curve.





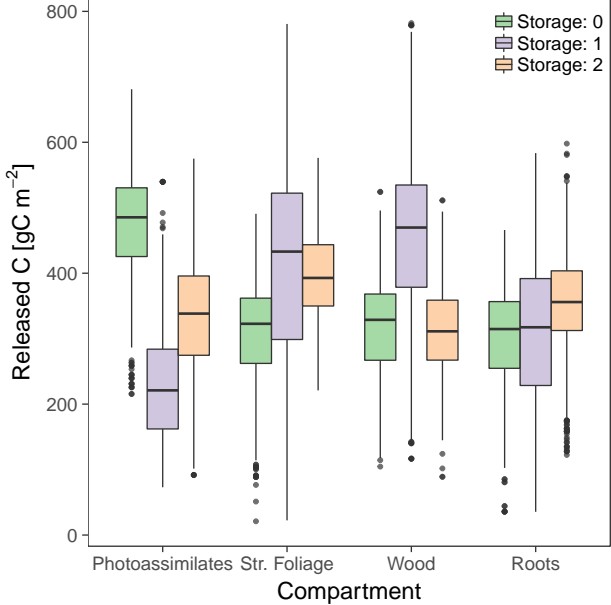

**Figure 5.** C release fluxes from the compartments at steady state, with uncertainty ranges obtained from the set of posterior parameters obtained by Bayesian optimization.

Differences in radiocarbon values for the different compartments hint to different levels of mixing of carbon fixed at different times. For the fast cycling compartments such as the Photoassimilates, the degree of mixing is relatively low because most of the radiocarbon reflects the values in the atmosphere. For other compartments that cycle at slower rates, the mix of recent and

5 old radiocarbon results in important divergences from the atmosphere. Mixing of carbon of different ages can be further studied with ages and transit distributions.

### 3.2.3 Age and transit time distributions

The age and transit time distributions were calculated assuming that the system was in steady state. These distributions had a wide range, expanding from 0 to several decades old carbon, and their shape varied according to the model structure (Figure 7).

Notice that distributions with the highest peak closer to 0 years, and with younger mean and median ages had the youngest C. Thus, the ascending order of the models according to their mean ages was: *Storage: 0*, *Storage: 1*, *Storage: 2*. As expected, the model with the oldest ages (*Storage: 2*) had the longest transit time. These trends were partially observed when we analyzed the uncertainties in mean age and mean transit times (Figure 7 (C) and (D)), but the uncertainties for the model without storage were large. The large uncertainties regarding the mean ages and transit times of the model *Storage: 0* might have resulted from

the high correlation between its parameters.

The above mentioned age-dependent ranking of the models only holds true for Wood (Figure 8), which was the compartment with the closest resemblance to the overall system age densities because it comprised most of the mass in the system. The





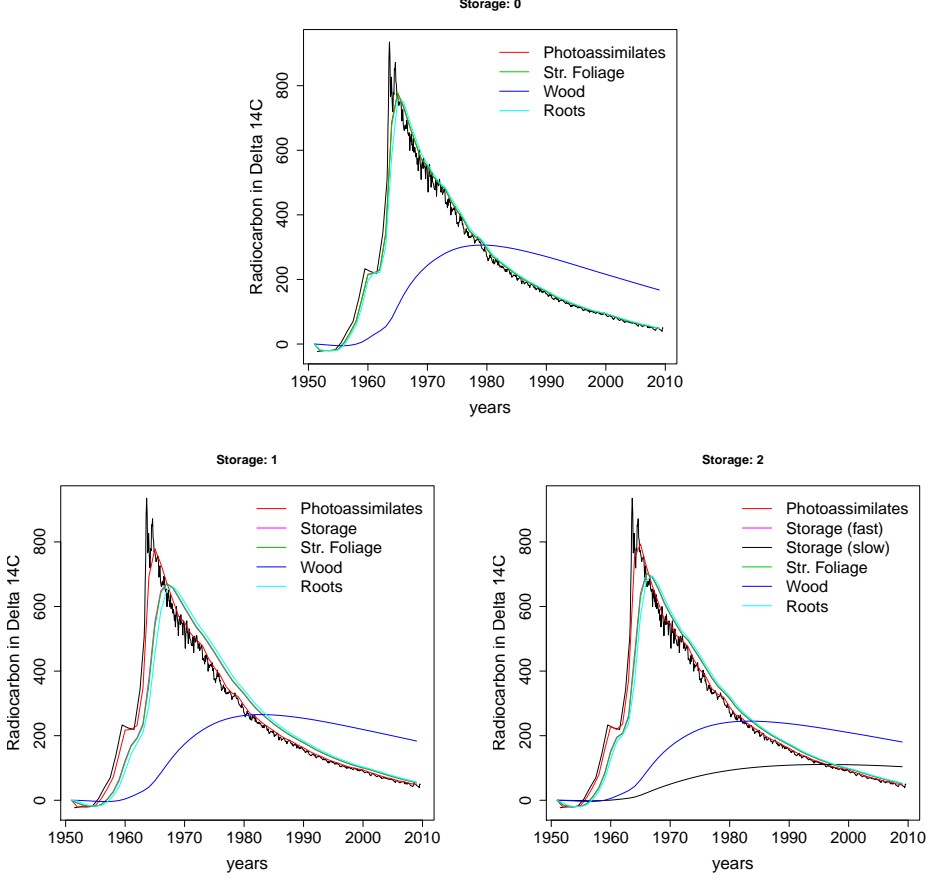

**Figure 6.** Radiocarbon simulations for the three model structures. The black curve corresponds to the $\Delta^{14}$C in the atmosphere, and the other colors depict the vegetation compartments.

inclusion of two storage compartments in *Storage: 2* resulted in a relatively 'flat' distribution; with a peak at very young ages, but with a long tail that lead to a mean age 10-20 years older than the other two models. This contrasts with the other models, which peaked at around the same time but had steeper curves.

5     The only compartment that had an age maximum at 0 years was the Photoassimilates. Hence, this compartment had a unique distribution curve reflecting the fact that all new carbon (age = 0 years) enters the models only through this compartment and it is later transferred to the others. Although the other fast cycling compartments (Str. Foliage and Roots) had peaks after 0 years, their C age was distributed towards young ages, with mean ages between 1 and 3 years. This spread in the C age of Str. Foliage for the models with storage may suggest either that the cycling rate of this compartment was relatively slow or that it received C from compartments with older carbon.

    The age densities of the storage compartments, just as the ones for Foliage, Wood, and Roots, consisted of curves with peaks at young ages and long tails (Figure 9). In the case of the fast cycling compartments, the mean age of the models *Storage: 1*



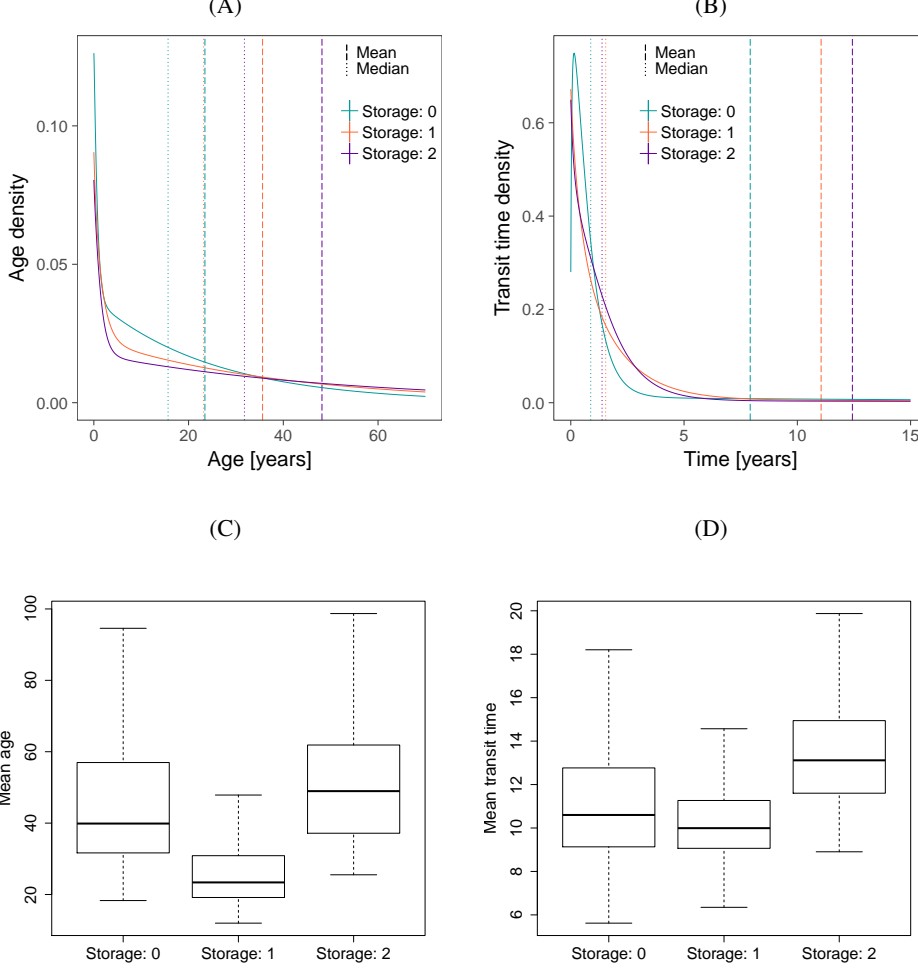

**Figure 7.** System ages and transit times. (A) Age and transit (B) time density distributions calculated for each of model structure using the best parameter set from the optimization; the dashed and dotted lines mark the mean and median ages, respectively. (C) Spread of mean ages (left) and mean transit times (D) obtained from all posterior parameter sets from the Bayesian optimization.

and *Storage: 2* was 1.71 and 2.14 years, respectively, but the long tail indicates that it is also probable to find 5-year-old C in this compartment. The mean age of the slow cycling compartment was 45 years, but the mixing of ages is also observed in the density curve, in which the age of C ranged from 0 to more than 150 years.





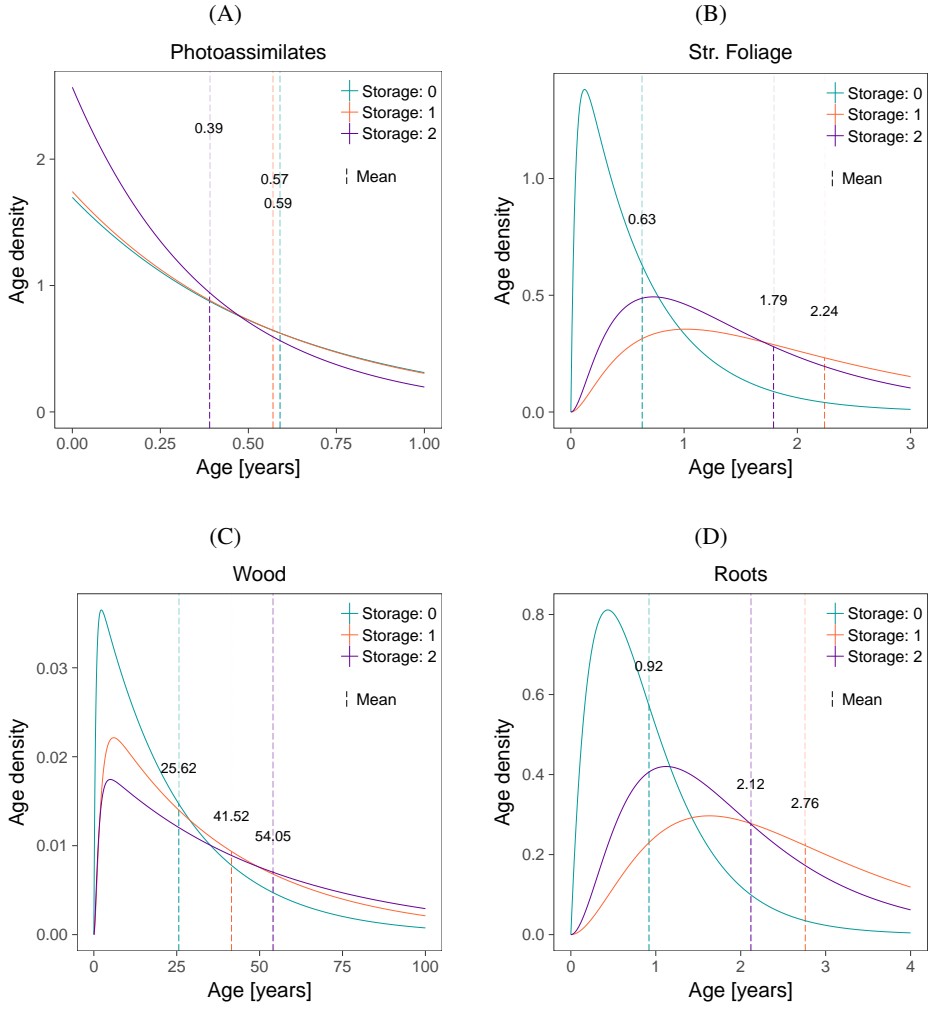

**Figure 8.** Age densities simulated for the compartments: (A) Photoassimilates, (B) Str. Foliage, (C) Wood and (D) Roots. Each model structure is depicted in a different color. Dashed lines correspond to mean ages.

## 4   Discussion

Our simulation results showed that C cycling in ecosystems can be largely influenced by different carbon allocation strategies, which may result in diverging carbon cycling predictions for specific simulations. However, not all of the different prediction
5   metrics were impacted with the same strength by the assumed number of compartments and values of cycling rates, so results here need to be interpreted within the context of predicted uncertainties.





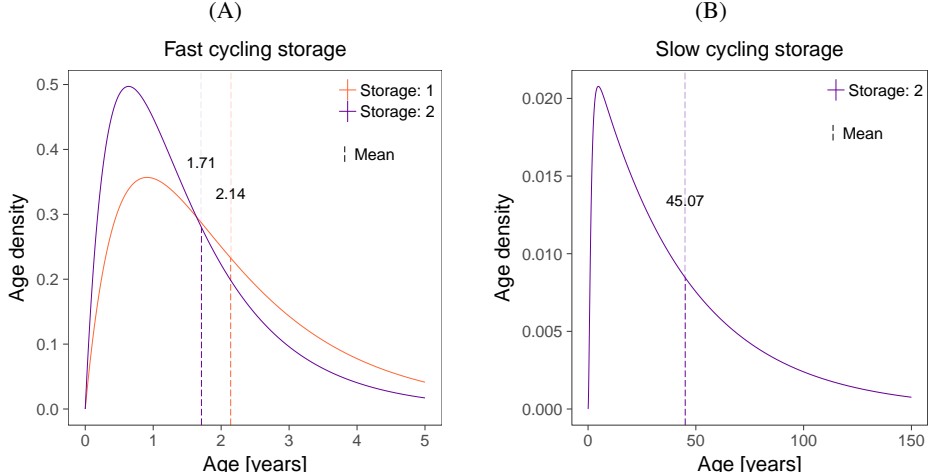

**Figure 9.** Age densities simulated for the models with storage compartments. (A) Fast cycling compartment of models *Storage: 1* and *Storage: 2*. (B) Slow cycling storage of the only model with 2 storage compartments.

### 4.1 Diagnosing model performance with C release fluxes, and age and transit time distributions

The simulated ecosystem properties that were more strongly impacted by the assumptions behind model structure were: the fluxes of C released from each compartment, and the ages and transit time distributions of carbon in the system and in each compartment. This sensitivity to different carbon allocation strategies makes them good candidates for diagnosing model performance.

Given that the C release fluxes from Photoassimilates, Str. Foliage, and Wood were highly sensitive to the three model structures, empirical measurements of these compartments could be used as constraints during the parameter estimation procedure. This is congruent with previous studies performed at the same site, where the use of data regarding woody biomass increment and LAI reduced the uncertainties in the predictions of net C sequestration and foliage dynamics, respectively (Keenan et al., 2013). Wood stock observations had also been proven to constrain well simulations of fine root mass (Smallman et al., 2017).

Despite less sensitivity to the different carbon allocation schemes, the radiocarbon accumulation was useful to diagnose the models' performance according to the cycling speed of their compartments. As an example, we could identify the short delay of the $\Delta^{14}$C signature of the Str. Foliage compartment with respect to the current year's atmosphere, in the models with storage (Figure 6). This means that this compartment had a slightly slower C cycling than what is expected for a deciduous forest (Trumbore et al., 2002, 2015). This delay in radiocarbon accumulation was also observed as shifts to older ages in the age

distributions (Figure 8), with a resolution of months. In general, radiocarbon measurements can also help to constrain model parameters with respect to how fast or slow different compartments cycle C (Trumbore et al., 2016), but with less resolution than the age and transit time distributions.



Overall, the age and transit distributions were the best candidates to diagnose model performance and potentially constrain parameter estimations, because they were the most sensible to differences in model structure and parameter values. So, what

had the highest impact on these distributions, the differences in cycling rates or the inclusion of storage compartments? It seems like these two characteristics had direct and indirect effects on the predictions of the above mentioned distributions, respectively.

On the one hand, as we initially inferred from the equations 2-5, the calculation of age and transit time distributions depends on the C partitioning schemes ($\beta$) and the transfer -cycling- rates between plant compartments (**B**). Therefore, different pa-

rameter values that compose the vector $\beta$ and matrix **B** directly result in the different calculations of ages and transit times. This was tested by running the three models using the same parameter set; although they still differed in the number of storage compartments, the differences between the age and transit time distributions in the whole system and in the compartments was minimal (Figures A6 and A7). These similarities were also observed for the radiocarbon accumulation (Figure A5). The only exceptions to the above were the fast cycling compartments of the model *Storage: 0*, which were faster than in the other two

models.

On the other hand, model structure had an indirect effect on the predictions of age and transit time, most likely because the addition of storage compartments impacted the outcome of the parameter estimation and these different parameter values then lead to different age and transit time predictions. An illustration of this is the fit of the models to the data (Figure A4): Running the three models with the parameter set that gave the best fit for the model with 2 storage compartments hampered

the fit of the model with no storage compartments to the woody C measurements. Thus, although the system had an external C input that could never be depleted because it was assumed to be a constant flux, the parameter estimation accounted for extra compartments by modifying the cycling rates to optimize the fit of the models to the data.

As expected, systems with ages distributed towards older values also have older transit time distributions. In fact their correlation can be confirmed by observing the formulas once again. The calculation of these two properties depends on the

matrix of transfer coefficients (**B**), but there are other factors driving these two distributions. The additional factor driving the mean age calculation is the relative amount of C stock at steady state; whereas the mean transit times depend on the C partitioning schemes ($\beta$) (Metzler and Sierra, 2017). So, the mean transit time calculation is only limited by the rates of C transfer, while the mean age of C in the system also depends on its mass. This is why the mean age of C in vegetation is determined by the age of the compartment where the majority of the mass is stored, which in this case is Wood. We further

explored this relation in the scatter plot in (Figure A9), where the distribution of the points below the 1:1 line indicate that the three models have mean ages greater than their mean transit times. This means that they have big masses of old carbon, but they also have highly dynamic compartments through which carbon transits very fast.

We also diagnosed the model performance by comparing the predicted ages for the storage compartments (Figure 9) with the mean age of NSC measured in previous empirical studies (Richardson et al., 2013). The mean age of NSCs from red maple

cores obtained at Harvard forest, was $7.2 \pm 7.7$ yr; for the fast cycling storage compartments of the models *Storage: 1* and *Storage: 2*, it was 1.71 and 2.14 yr respectively, and 45 yr for the slow cycling storage. Clearly, the mean ages of fast storage compartments are smaller than the mean value calculated empirically, but given the uncertainty of the measurements, they are



still under the observed range. Furthermore, the density distributions of these storage compartments show that even though they are well-mixed, their C do not have the same age. Thus, it is also likely to find C between 0-5 yr and 0-150 yr in the
fast and slow cycling storage compartments, respectively. This resonates with the fact that although stemwood NSC is highly dynamic on seasonal time-scales, it can also be surprisingly old (Richardson et al., 2013). Then, the two hypothesis regarding C mixing -inward mixing of younger and older C in one compartment (Trumbore et al., 2015), and 2 compartments (young and old) that mix (Richardson et al., 2013)- converge in the concept of age distributions because C is simultaneously been fixed and removed from the compartments at different times, a process that results in C age distributions. We can think about these
dynamics in the context of a stochastic process. The total amount of C that enters and leaves each compartment is fixed and given by the deterministic model, but the time that each C particle stays in a compartment varies randomly within them. So, the age distribution of C particles in each compartment is a mix of new and old carbon, with distribution functions emerging from the deterministic model (equation 2).

Another important observation regarding the mean age predictions is the fact that these calculations were performed under
the assumption that the system, in this case the forest, was in steady state. Since the C stocks in Harvard forest continue growing, the calculated mean ages and transit times should be interpreted as predictions of the mean age that the carbon may have in this forest once it is in steady state. Based on this, the time that this forest will take to reach the steady state is highly divergent among the three models. As an example, the model with two storage compartments would predict 20 yr more of growth to reach steady-state than the model with no compartments. If the environmental forcing was included in the simulations, the
mean ages and transit times would then have to be calculated using another set of equations (e.g., Rasmussen et al., 2016). However, to use such equations we need to know the history of inputs and cycling rates for the duration of the simulation, and that information was not available for Harvard forest. In any case, if there were external factors influencing the simulations, the predicted mean ages and transit times would be different in every time step.

It is also noteworthy that what we assume to be a compartment, e.g. Wood, does not necessarily meet the well mixed
assumption, so its particles may not have the same probability to leave the compartment at all times. Richardson et al. (2015) found a low concentration of old NSC in old rings of stemwood, and a high concentration of old NSC in coarse roots and fine roots of pine. Additionally, they found young and old C in roots. These dynamics were interpreted as poor C mixing and reserves that turn over on different timescales (Richardson et al., 2015), but they may also obey to physiological constraints; for example, the parenchyma in heartwood is thought to be dead, so NSC trapped in there may no longer be accessible to
the plant (Richardson et al., 2015), but it has some probability of being respired and lost from the system. So, to study the physiological significance of these findings with models, we might have to include such details regarding tree physiology. The important point we want to emphasize however, is that mixing of carbon in different vegetation compartments results in C age distributions that have been little studied previously. Our results are a first attempt to obtain these distributions using a number of assumptions, but more detailed models relaxing some of these assumptions would be needed to obtain more accurate C age
distributions.

Although there are still knowledge gaps regarding plant physiology, and current C-dating methods only measure mean age of C rather than age distributions (Richardson et al., 2013), we expect our results to motivate future work, particularly in



the use of isotope tracers and their time evolution to approximate age distributions. Biosphere models can be enhanced with structural adjustments and the uncertainties in the parameter values can be reduced by constraining them with age and transit time distributions. These improved models could then be used to test hypothesis regarding physiological questions, and assess the sustainability of current terrestrial C sinks, given changes in environmental forcing.

## 4.2 Model equifinality (identifiability)

Model equifinality (Medlyn et al., 2005) was evident from the fact that despite having different number of compartments and values of cycling rates, all of the three models had similar simulations of C stocks that fitted the data well (Figures 3 and 4). Along with model equifinality, we obtained a high collinearity between some parameters, implying that they are non-identifiable, i.e., they cannot be uniquely estimated from the given data sets (Soetaert and Petzoldt, 2010). Thus, for these particular models, the time-course measurements from only 2 out of 4-6 vegetation compartments is not sufficient to estimate

the values of 7-10 parameters.

Model equifinality as well as the impossibility to uniquely identify certain parameters (parameter non-identifiability) may be the result of the high correlations between the parameters. Positive parameter correlations, may indicate a *practical non-identifiability*, where the insufficiency or poor quality of data is not a good constrain for the parameters. In addition, a negative parameter correlation can be a symptom of *structural non-identifiability*, which is the result of a redundant parameterization

(Raue et al., 2009; Timmer, 2011; Raue et al., 2012; Cressie et al., 2009). Thus, the three models had practical and structural non-identifiabilities, which means that they need to be constrained with more and better data, and they need to be restructured in order to avoid compensation of fluxes into and out of the compartments.

Since this study was limited by the availability of relevant empirical data, the parameter values that we used are only one of many possible outcomes of parameter estimations using the same data sets. Therefore, it is possible that none of these models

accurately depict the C cycle in the Harvard Forest. However, these problems experienced with parameter non-identifiability are not an isolated case; the process of finding unknown rates of C sequestration by fitting biosphere models to empirical data (Luo et al., 2003) is often hampered by the parameter non-identifiability (Schaber and Klipp, 2011). This is a real problem because parameters such as those that correspond to carbon turnover explain most of the variation in the response of terrestrial vegetation to future climate and $CO_2$ (Friend et al., 2014) and are highly important in determining C age and transit times.

Nevertheless, we succeeded in assessing the influence of model structure in predicted ecosystem processes.

## 5 Conclusions

We obtained age and transit times distributions of carbon for simple vegetation models with contrasting carbon allocation schemes. Our results show that mixing of carbon in different vegetation compartments results in C age distributions not explored before in previous studies. The shape of these distributions depends largely on model structure, and in particular on

how carbon allocation is represented in models. While the inclusion of a carbon storage compartment had a small impact on





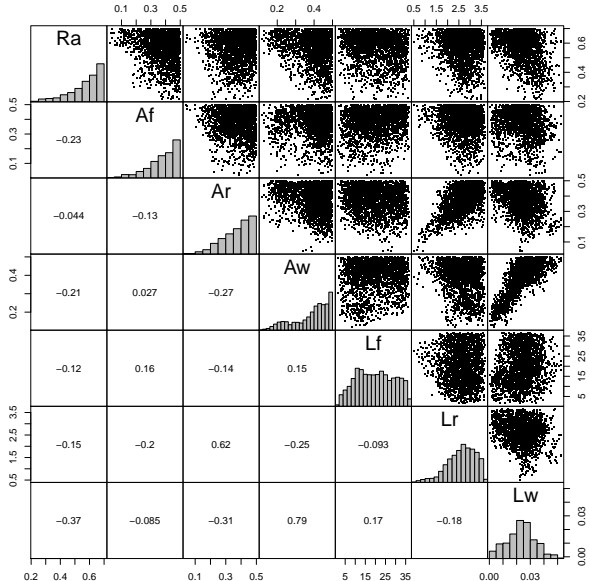

**Figure A1.** Pairwise plots of sensitivity functions for the model *Storage: 0*

predicting C stocks in vegetation compartments, it resulted in very different predictions of age, transit time distributions, C release, and isotopic composition.

Models with none or one storage compartment may fail to explain the mixing of ages found in different vegetation compartments, but they are more parsimonious than the model with 2 storage compartments. Nonetheless, parameter collinearity and model equifinality were persistent problems that might be solved if more constraints are added, since the time series of C in foliage and wood are not enough to parameterize a full vegetation model.

Our results suggest that ages and transit times, which can be indirectly measured using isotope tracers, can be used to
5  improve biosphere models via examination of their structure and estimation of parameter values, which then can be used to assess the strength of C sources or sinks from vegetation.

Finally, it is advantageous to consider age and transit times as distributions, rather than only mean values; with their distributions we obtain additional insights on the temporal dynamics of carbon use, storage, and allocation, which not only depends on the rate at which C flows into and out of the compartments, but also on the stochastic nature of the process itself.

## Appendix A:  Appendix

*Competing interests.*  The authors declare that they have no conflict of interest.





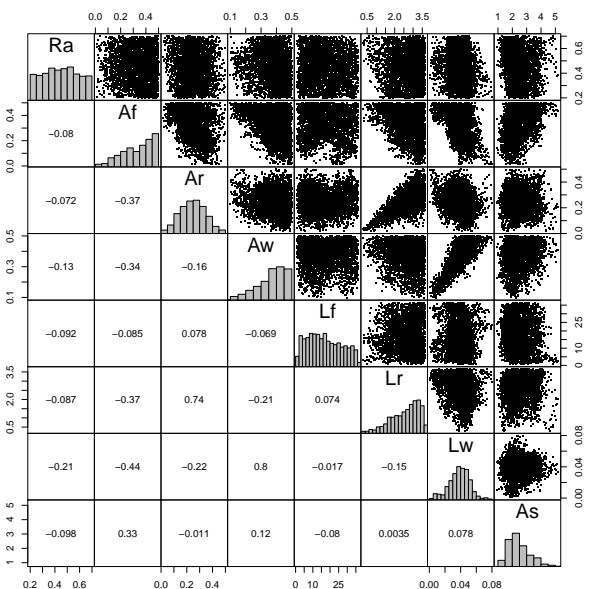

**Figure A2.** Pairwise plots of sensitivity functions for the model *Storage: 1*

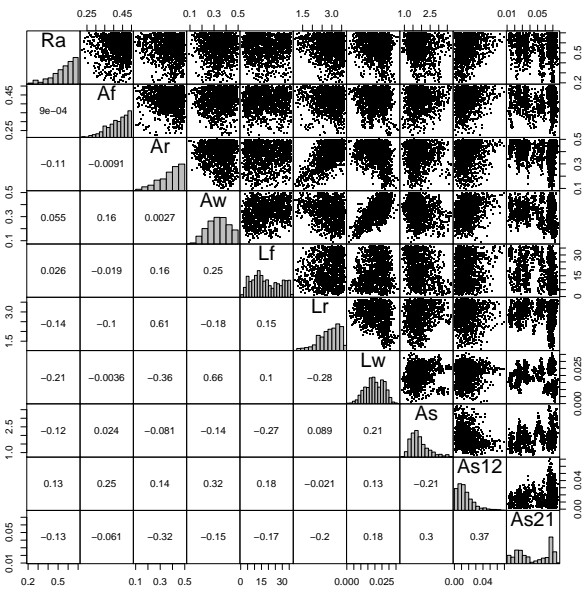

**Figure A3.** Pairwise plots of sensitivity functions for the model *Storage: 2*



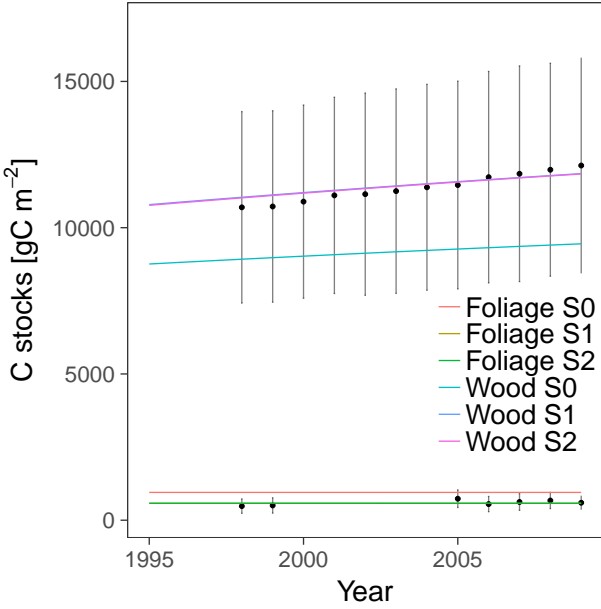

**Figure A4.** Carbon stocks estimated for each model. The three models were run using the same parameter set.

*Acknowledgements.* Research at Harvard Forest is supported by the National Science Foundation's LTER program (DEB-1237491). This material is based upon work supported by the U.S. Department of Energy, Office of Science, Office of Biological and Environmental Research. Part of this work was the result of a research visit to the Terrestrial Ecosystems and Global Change group, Department of Organismic and Evolutionary Biology, Harvard University. It was funded by the Max Planck Society and the German Research Foundation through its
Emmy Noether Program (SI 1953/2–1).





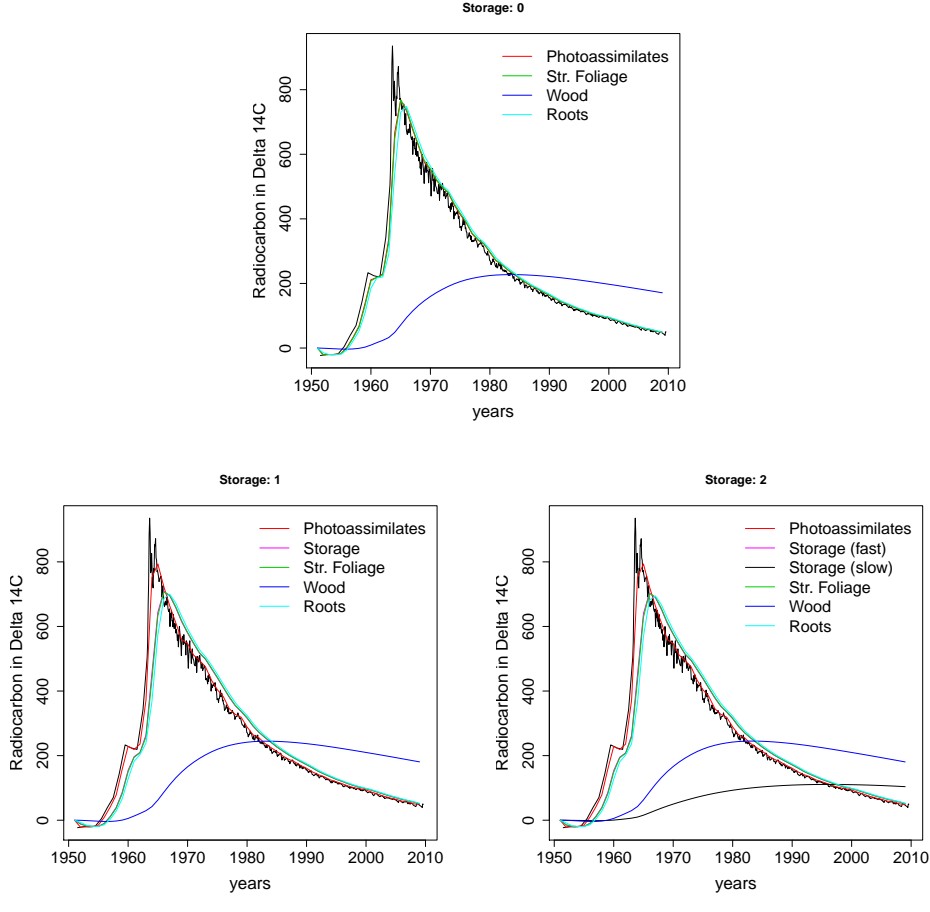

**Figure A5.** Radiocarbon simulations for the three models. The three models were run using the same parameter set. The black curve corresponds to the $\Delta^{14}C$ accumulation in the atmosphere, and the other colors depict the vegetation compartments.

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




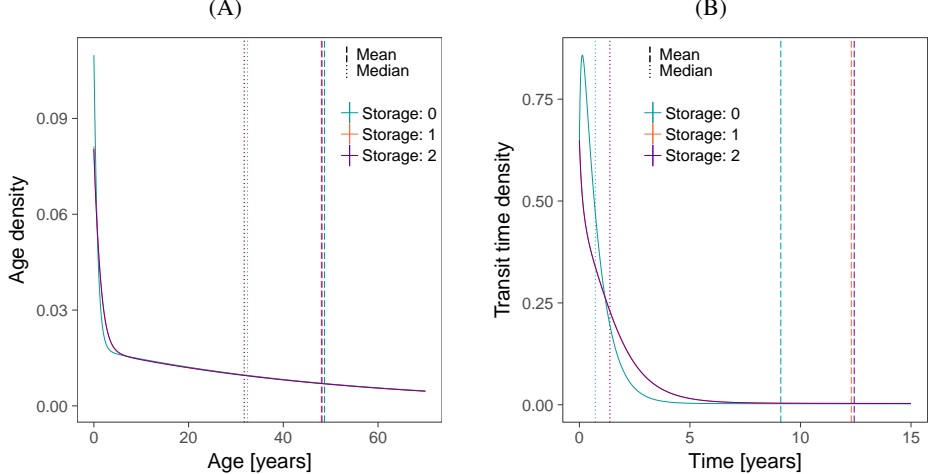

**Figure A6.** System ages and transit times. (A) Age and transit time (B) distributions calculated for each of the three models using the same parameter set; the dashed and dotted lines mark the mean and median ages, respectively.

Fox, A., Williams, M., Richardson, A. D., Cameron, D., Gove, J. H., Quaife, T., Ricciuto, D., Reichstein, M., Tomelleri, E., Trudinger, C. M., and Wijk, M. T. V.: The REFLEX project: Comparing different algorithms and implementations for the inversion of a terrestrial ecosystem model against eddy covariance data, Agricultural and Forest Meteorology, 149, 1597 – 1615, doi:http://dx.doi.org/10.1016/j.agrformet.2009.05.002, 2009.

Friedlingstein, P., Cox, P., Betts, R., Bopp, L., Von Bloh, W., Brovkin, V., Cadule, P., Doney, S., Eby, M., Fung, I., Bala, G., John, J., Jones,
5    C., Joos, F., Kato, T., Kawamiya, M., Knorr, W., Lindsay, K., Matthews, H. D., Raddatz, T., Rayner, P., Reick, C., Roeckner, E., Schnitzler, K. G., Schnur, R., Strassmann, K., Weaver, A. J., Yoshikawa, C., and Zeng, N.: Climate-Carbon Cycle Feedback Analysis: Results from the $C^4$MIP Model Intercomparison, Journal of Climate, 19, 3337–3353, 2006.

Friend, A. D., Lucht, W., Rademacher, T. T., Keribin, R., Betts, R., Cadule, P., Ciais, P., Clark, D. B., Dankers, R., Falloon, P. D., Ito, A., Kahana, R., Kleidon, A., Lomas, M. R., Nishina, K., Ostberg, S., Pavlick, R., Peylin, P., Schaphoff, S., Vuichard, N., Warszawski, L., Wiltshire, A., and Woodward, F. I.: Carbon residence time dominates uncertainty in terrestrial vegetation responses to future climate and atmospheric $CO_2$., Proceedings of the National Academy of Sciences of the United States of America, 111, 3280–5, doi:10.1073/pnas.1222477110, 2014.

5  Grulke, N. E., Andersen, C. P., and Hogsett, W. E.: Seasonal changes in above- and belowground carbohydrate concentrations of ponderosa pine along a pollution gradient, Tree Physiology, 21, 173, doi:10.1093/treephys/21.2-3.173, 2001.

Hartmann, H. and Trumbore, S.: Understanding the roles of nonstructural carbohydrates in forest trees -from what we can measure to what we want to know, New Phytologist, 211, 386–403, doi:10.1111/nph.13955, 2016.

Hartmann, H., Ziegler, W., and Trumbore, S. E.: Lethal drought leads to reduction in nonstructural carbohydrates in Norway spruce tree roots but not in the canopy, Functional Ecology, 27, 413–427, doi:10.1111/1365-2435.12046, 2013.

Hoch, G. and Körner, C.: The carbon charging of pines at the climatic treeline: a global comparison, Oecologia, 135, 10–21,
doi:10.1007/s00442-002-1154-7, 2003.



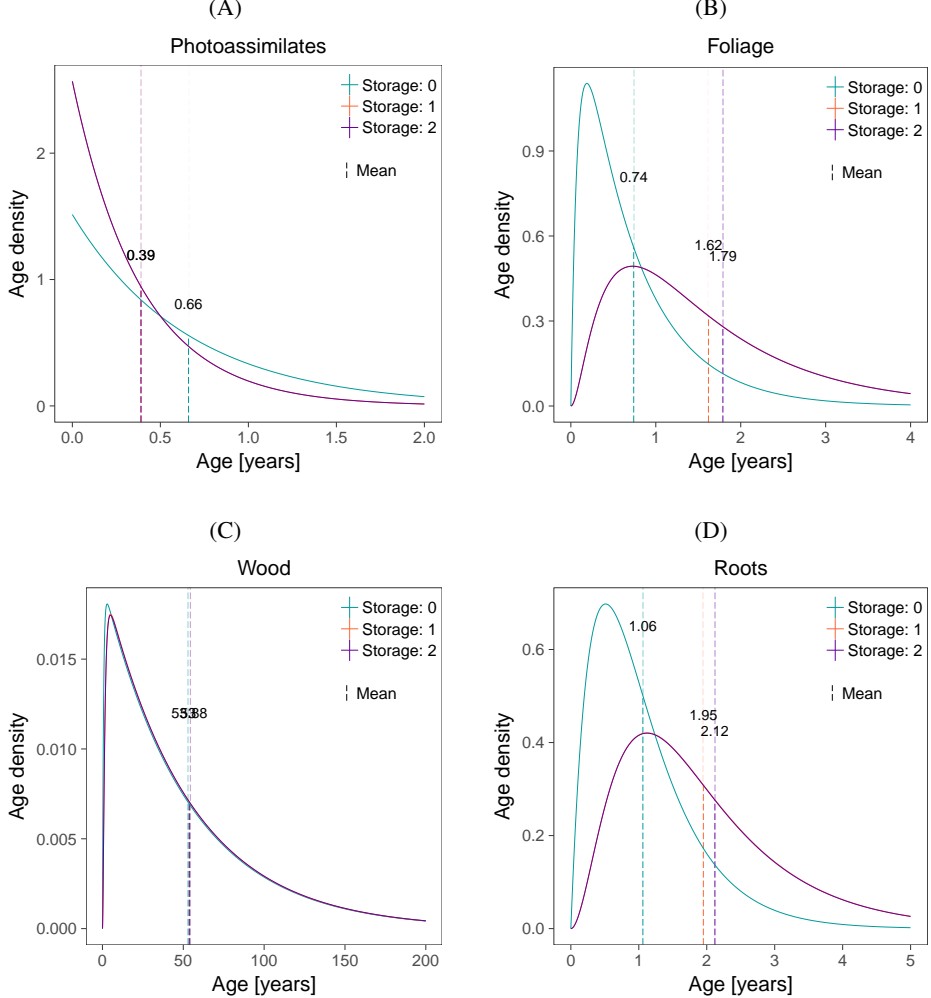

**Figure A7.** Age densities simulated for the compartments: (A) Photoassimilates, (B) Str. Foliage, (C) Wood and (D) Roots. The three models were run using the same parameter set. Each model is depicted in a different color. The dashed lines correspond to the mean ages of each model for each compartment.

Keenan, T. F., Davidson, E. A., Munger, J. W., and Richardson, A. D.: Rate my data: quantifying the value of ecological data for the development of models of the terrestrial carbon cycle, Ecological Applications, 23, 273–286, doi:10.1890/12-0747.1, 2013.

Körner, C.: A matter of tree longevity, Science, 355, 130–131, doi:10.1126/science.aal2449, 2017.

Lacointe, A.: Carbon allocation among tree organs: A review of basic processes and representation in functional-structural tree models, Annals of Forest Science, 57, 521–533, doi:10.1051/forest:2000139, 2000.

Luo, Y., White, L. W., Canadell, J. G., DeLucia, E. H., Ellsworth, D. S., Finzi, A., Lichter, J., and Schlesinger, W. H.: Sustainability of terrestrial carbon sequestration: A case study in Duke Forest with inversion approach, Global Biogeochemical Cycles, 17, n/a–n/a, doi:10.1029/2002GB001923, 2003.



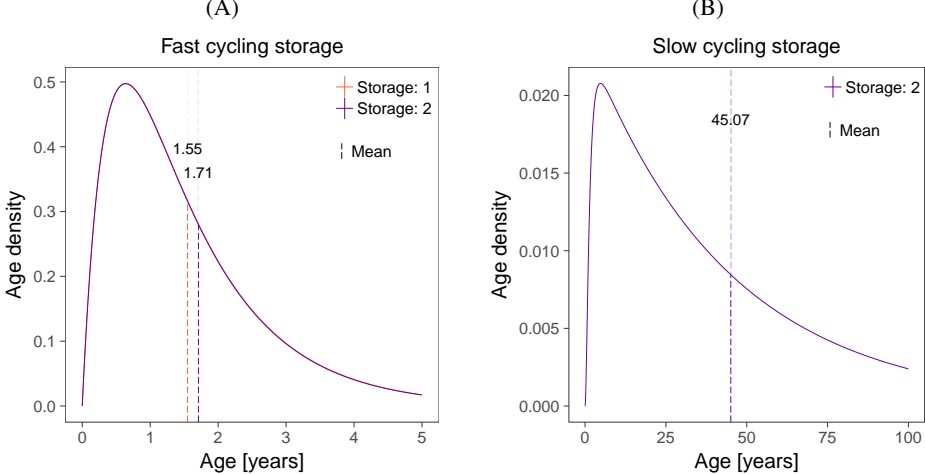

**Figure A8.** Age densities simulated for the models with storage compartments; the models were run using the same parameter set. (A) Fast cycling compartment of models *Storage: 1* and *Storage: 2*. (B) Slow cycling storage of the only model with 2 storage compartments.

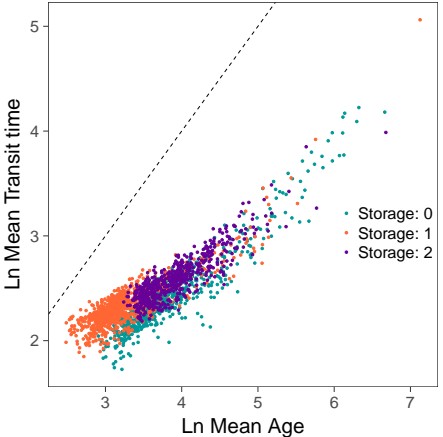

**Figure A9.** Scatter plot of mean age Vs. mean transit times in log scale. The three models have distributions below the 1:1 line.

Luo, Y., Shi, Z., Lu, X., Xia, J., Liang, J., Jiang, J., Wang, Y., Smith, M. J., Jiang, L., Ahlström, A., Chen, B., Hararuk, O., Hastings, A.,

Hoffman, F., Medlyn, B., Niu, S., Rasmussen, M., Todd-Brown, K., and Wang, Y.-P.: Transient dynamics of terrestrial carbon storage: mathematical foundation and its applications, Biogeosciences, 14, 145–161, doi:10.5194/bg-14-145-2017, 2017.

Martínez-Vilalta, J., Sala, A., Asensio, D., Galiano, L., Hoch, G., Palacio, S., Piper, F. I., and Lloret, F.: Dynamics of non-structural carbo-hydrates in terrestrial plants: a global synthesis, Ecological Monographs, 86, 495–516, doi:10.1002/ecm.1231, 2016.

Medlyn, B. E., Robinson, A. P., Clement, R., and McMurtrie, R. E.: On the validation of models of forest CO2 exchange using eddy

covariance data: some perils and pitfalls, Tree Physiology, 25, 839–857, doi:10.1093/treephys/25.7.839, 2005.




Metzler, H. and Sierra, C. A.: Linear Autonomous Compartmental Models as Continuous-Time Markov Chains: Transit-Time and Age Distributions, Mathematical Geosciences, doi:10.1007/s11004-017-9690-1, 2017.

Muhr, J., Messier, C., Delagrange, S., Trumbore, S., Xu, X., and Hartmann, H.: How fresh is maple syrup? Sugar maple trees mobilize carbon stored several years previously during early springtime sap-ascent, New Phytologist, 209, 1410–1416, doi:10.1111/nph.13782, 2016.

Rasmussen, M., Hastings, A., Smith, M. J., Agusto, F. B., Chen-Charpentier, B. M., Hoffman, F. M., Jiang, J., Todd-Brown, K. E. O., Wang, Y., Wang, Y.-P., and Luo, Y.: Transit times and mean ages for nonautonomous and autonomous compartmental systems, J. Math. Biol., 73, 1379–1398, doi:10.1007/s00285-016-0990-8, 2016.

Raue, A., Kreutz, C., Maiwald, T., Bachmann, J., Schilling, M., Klingmüller, U., and Timmer, J.: Structural and practical identifiability analysis of partially observed dynamical models by exploiting the profile likelihood, Bioinformatics, 25, 1923, doi:10.1093/bioinformatics/btp358, 2009.

Raue, A., Kreutz, C., Theis, F. J., and Timmer, J.: Joining forces of Bayesian and frequentist methodology: a study for inference in the presence of non-identifiability, Philosophical Transactions of the Royal Society of London A: Mathematical, Physical and Engineering Sciences, 371, doi:10.1098/rsta.2011.0544, 2012.

Richardson, A. D., Williams, M., Hollinger, D. Y., Moore, D. J. P., Dail, D. B., Davidson, E. A., Scott, N. A., Evans, R. S., Hughes, H., Lee, J. T., Rodrigues, C., and Savage, K.: Estimating parameters of a forest ecosystem C model with measurements of stocks and fluxes as joint constraints, Oecologia, 164, 25–40, doi:10.1007/s00442-010-1628-y, 2010.

Richardson, A. D., Carbone, M. S., Keenan, T. F., Czimczik, C. I., Hollinger, D. Y., Murakami, P., Schaberg, P. G., and Xu, X.: Seasonal dynamics and age of stemwood nonstructural carbohydrates in temperate forest trees, New Phytologist, 197, 850–861, doi:10.1111/nph.12042, 2013.

Richardson, A. D., Carbone, M. S., Huggett, B. A., Furze, M. E., Czimczik, C. I., Walker, J. C., Xu, X., Schaberg, P. G., and Murakami, P.: Distribution and mixing of old and new nonstructural carbon in two temperate trees, New Phytologist, 206, 590–597, doi:10.1111/nph.13273, 2015.

Schaber, J. and Klipp, E.: Model-based inference of biochemical parameters and dynamic properties of microbial signal transduction networks, Current Opinion in Biotechnology, 22, 109–116, doi:10.1016/j.copbio.2010.09.014, 2011.

Schiestl-Aalto, P., Kulmala, L., Mäkinen, H., Nikinmaa, E., and Mäkelä, A.: CASSIA – a dynamic model for predicting intra-annual sink demand and interannual growth variation in Scots pine, New Phytologist, 206, 647–659, doi:10.1111/nph.13275, 2015.

Sierra, C. A., Müller, M., and Trumbore, S. E.: Models of soil organic matter decomposition: The SoilR package, version 1.0, Geoscientific Model Development, 5, 1045–1060, doi:10.5194/gmd-5-1045-2012, 2012.

Sierra, C. A., Müller, M., Metzler, H., Manzoni, S., and Trumbore, S. E.: The muddle of ages, turnover, transit, and residence times in the carbon cycle, Global Change Biology, in print, doi:10.1111/gcb.13556, 2016.

Smallman, T. L., Exbrayat, J. F., Mencuccini, M., Bloom, A. A., and Williams, M.: Assimilation of repeated woody biomass observations constrains decadal ecosystem carbon cycle uncertainty in aggrading forests, Journal of Geophysical Research: Biogeosciences, doi:10.1002/2016JG003520, 2017.

Soetaert, K. and Petzoldt, T.: Inverse Modelling, Sensitivity and Monte Carlo Analysis in R Using Package FME, Journal of Statistical Software, 33, doi:10.18637/jss.v033.i03, 2010.

Timmer, J.: Addressing parameter identifiability by model-based experimentation, IET Systems Biology, 5, 120–130(10), 2011.

Trumbore, S., Gaudinski, J. B., Hanson, P. J., and Southon, J. R.: Quantifying ecosystem-atmosphere carbon exchange with a 14C label, Eos, Transactions American Geophysical Union, 83, 265–268, doi:10.1029/2002EO000187, 2002.





Trumbore, S., Czimczik, C. I., Sierra, C. A., Muhr, J., and Xu, X.: Non-structural carbon dynamics and allocation relate to growth rate and leaf habit in California oaks, Tree Physiology, 35, 1206, doi:10.1093/treephys/tpv097, 2015.

Trumbore, S. E., Sierra, C. A., and Hicks Pries, C. E.: Radiocarbon and Climate Change: Mechanisms, Applications and Laboratory Techniques, chap. Radiocarbon Nomenclature, Theory, Models, and Interpretation: Measuring Age, Determining Cycling Rates, and Tracing
Source Pools, pp. 45–82, Springer International Publishing, doi:10.1007/978-3-319-25643-6_3, 2016.

460   Urbanski, S., Barford, C., Wofsy, S., Kucharik, C., Pyle, E., Budney, J., McKain, K., Fitzjarrald, D., Czikowsky, M., and Munger, J. W.: Factors controlling CO2 exchange on timescales from hourly to decadal at Harvard Forest, Journal of Geophysical Research: Biogeosciences (2005–2012), 112, doi:10.1029/2006JG000293, 2007.

Wofsy, S., Goulden, M., Munger, J., Fan, S.-M., Bakwin, P., Daube, B., Bassow, S., and Bazzaz, F.: Net exchange of CO2 in a mid-latitude forest, Science, 260, 1314–1317, 1993.

465   Xia, J., Luo, Y., Wang, Y. P., and Hararuk, O.: Traceable components of terrestrial carbon storage capacity in biogeochemical models, Global Change Biology, 19, 2104–2116, doi:10.1111/gcb.12172, 2013.

Yizhao, C., Jianyang, X., Zhengguo, S., Jianlong, L., Yiqi, L., Chengcheng, G., and Zhaoqi, W.: The role of residence time in diagnostic models of global carbon storage capacity: model decomposition based on a traceable scheme, Scientific Reports, 5, doi:10.1038/srep16155, 2015.