# Peer review of "Ages and transit times as important diagnostics of model performance for predicting carbon dynamics in terrestrial vegetation models"

_Biogeosciences, 2017_

## Referee Comment (RC1) · Anonymous Referee #1 · 11 Sep 2017

General comments : The research article 'Ages and transit times as important diagnostics of model performance for predicting carbon dynamics in terrestrial vegetation models' promotes using the distribution of C ages and C transit times of different tree organs for improving performance of vegetation model. In order to do so, the authors tested three different carbon allocation schemes into a simple vegetation model with the aim to discriminate the three allocation models in terms of C stock, C flux, radiocarbon, C ages and C transit times distribution. The paper concludes that C ages and C transit times distribution can indeed be used to evaluate the skill of the allocation

schemes. Furthermore, the authors encourage the scientific community to use their approach for future model comparison or validation. In my opinion the authors did a good job in developing a powerful method to help reducing uncertainty in model output as well as facilitating model development. I really appreciate that the R scripts are clear and easy to use by someone interested in applying this method to their own model. I'm convinced that the ideas presented in this paper our of interest to the readership of biogeosciences, but the presentation itself needs to be improved. The methods and result sections lack essential information. Before resubmitting the manuscript, a senior researcher should carefully edit these sections such that they meet the minimal requirements for publication. Furthermore, the following concerns should be addressed in the manuscript:

1) The study relies on theoretical simulations from an unrealistic vegetation model to match the requirements of the mathematical calculation of ages and transit time. I understand and agree with the need for this approach. This caveat should, however, be addressed in the discussion and the conclusion where it was argued that the method can be useful for evaluating more complex vegetation models. In my opinion this suggestion is overly simple and this statement is not needed for the study. I would recommend the authors to stick to the more theoretical conclusions. Based on my understanding of complex vegetation models, the numerical and conceptual work required to implement the proposed method into a vegetation model with full complexity is too large to be downplayed. If the authors disagree, I would expect at least a paragraph dedicated to how this functionality couldbe integrated into a model with phenology, environmental changes, forest management i.e. forest structure and genetics. This would really enhance the impact of the manuscript as it may convince modellers to apply your method in their models.

2) The method section is concise, which in general I like, but the current description of the methods is too concise making it hard to follow. Each time the authors refer to a method described by another study they referring to it without a explanation or

at least the reason why they choose this one. As a general guideline readers should be able to undertand the method section without having to consult any other papers. What justifies the use of a two-step instead of one step optimization? Why do you need Bayesian optimization? Why do you test co-linearity between parameters (answered too late in the discussion) ? 3) Since the authors use a lot of R packages to manage complex statistical methods, I guess that they are familiar with statistics and coding. Surprisingly, simple tests such as the t-test, anova, or metrisc like Root Mean Squared Error are not used to highlight their finding. The conclusions are not backed-up by any statistics, which is mandatory for publication in a journal like Biogeosciences.

Detailed comments on the tables and figures : Table 1: the column called 'final' is not clear ! Why this column differs from 'Best2' column ? I guess it is because of the constraint optimization on C stock but it is not described in the caption. Do not show the median and the quantiles of Best2 if you will not use these values in the next step.

Table 2:I am not sure this one is necessary ...

Figure 3: Two figures instead of one can be better to avoid the compressing of the leaf C stock.

Figure 5,7: Put some statistics to highlight your results. You argue that mean ages and transit time are different across allocation model. Support it with a probability!

Figure 8: Again show that mean ages are significantly different.

---

## Referee Comment (RC2) · Anonymous Referee #2 · 11 Sep 2017

Despite an encouraging title and a promising abstract, I find the study led by Ceballos-Núñez & al. quite disappointing and I doubt that the modelling approach proposed here can be used by a larger community to constrain carbon dynamics in terrestrial vegetation model.

The "vegetation" model described in this study is in fact a very simple box model, calculating fluxes between different carbon storage compartments, as well as the carbon stock of each compartment, comparing three model structures (i.e. increasing the number of carbon pools). The model was forced by a constant input of carbon (GPP=1400

gC m-2 year-1) and run on a yearly time step, with no change in environmental forcing (climate, CO2,etc.). The results are shown during the transient spin-up (e.g. fig4) or at steady state (most figures).

In my opinion, this approach (i.e. yearly time step, constant GPP, no external forcing) is absolutely not appropriate for "predicting carbon dynamic" as claimed in the title. The actual dynamic of the carbon cycle, i.e. the increasing terrestrial carbon sink, is happening because of transient changes in external environmental conditions affecting the terrestrial carbon dynamics. Therefore it is impossible to draw any conclusion from this study, with respect to the actual dynamic of the system.

I also disagree on the way the authors claim they evaluated their model using observation. [i.e. 'We found a good fit of the three model structures to the available data' [abstract]]. First, I understand that the data used for model "evaluation" are the same that the one used for model optimisation as these are the only data mentioned in the manuscript. Is the model calibrated against Harvard Forest data (i.e. results in Table 1) and then compared to the same data for evaluation (Figures 3 and 4)? Or did I miss something? Second, there is no information on how the simulations were done for evaluation. Figure 4 clearly shows that the model is in transient conditions from 1950 to 2010, with wood carbon stocks increasing and being comparable to the observations in 2010. That would make some sense if the model actually started in 1950, with external forcing (climate, CO2, land use, etc.) changing from year to year. My understanding is that this is not the case here. The model is simply spinning up, slowly reaching steady state. The agreement in 2010 is hence completely artificial. Cstock wood does not seem the have reached equilibrium, running another 100 years and it would be well above the observations. Unless I missed these two elements, there is strictly no evaluation in this paper.

Finally the co-authors conclude that 'Differences in model structures had a small impact on predicting C stocks in ecosystem compartments, but overall they resulted in very different predictions of age and transit time distributions'. I will argue here that

considering the fact that each of their model parameters was constrained using the same carbon stocks, this is a result one should expect as a direct outcome of their methodology and no conclusion on the reality of the processes can be drawn from it.
* * *

---

## Referee Comment (RC3) · Anonymous Referee #3 · 11 Sep 2017

The manuscript by Ceballos-Núñez et al. describes a study on transit times of C through a temperate forest, with the help of a framework with a number of pools and transfer time coefficients between these pools. The number of non-structural pools is varied to investigate the importance of these for representing the transfer.

Whereas the concept used in the manuscript is generally well explained, more information should be provided on the details. I have some reservations to the chosen methods, but this may also be partly caused by misunderstanding due to the too concise description. Some parts of the study need to be clarified, and slight alterations to

the methodology may be required.

This manuscript can become an interesting contribution to the literature on modelling carbon allocation in (temperate forest) ecosystems, but I would recommend the authors to clarify the methodology, and also to rethink some of the concepts that they apply in their setup. I hope that the comments below can support the authors in revising their work.

Major remarks

A general issue with the methodology is the limited amount of observations that is used to constrain the transfer matrix B. Firstly, these data should be described more carefully, e.g., Figures 3 and 4 seem to suggest that wood or biomass numbers for multiple years are available, which does not appear in the Methods section. Secondly, and more importantly, it seems to me that the available data make it impossible to separate the non-structural pool(s) in the models, which leaves the system underdetermined. The authors also seem to refer to this in their methods (p 7/l 5). It is unclear how a Monte-Carlo setup for sampling the parameter space could help to constrain these, and in fact the authors seem to suggest that it does not. I would suggest to either explain more carefully how the Monte-Carlo setup was applied, which parameters were estimated with it, and which constraints were used, or to skip the Monte-Carlo setup and replace it by a more simple parameter estimation based on literature values, which seems more appropriate for a system that cannot be constrained by observations.

Although the setup of the three steady-state models is generally well described in the methods, there is one aspect that I have not been able to resolve with regard to the photoassimilates pool. The text seems to suggest that this is not a real pool but just partitions the input vector u to the other pools and to autotrophic respiration, leaving it in essence a flux (or sum of fluxes) rather than a pool. However, in Fig. 2, $C_P$ receives its own box, and the presented results (e.g. Fig. 8) indicate that the photoassimilates pool also has an age. If this is indeed the case in the model, there would be one nonstructural pool (namely, C_P) in Storage 0, and two respectively three non-structural pools in Storage 1 and Storage 2. Maybe the authors can clarify whether this is the case or not. If it is, I would recommend to remove it as a pool, to reduce the size of the vector x and thereby reduce the number of parameters that need to be estimated.

The discussion section describes the limitations of the method to some extent, but it should also be used to discuss the validity of the assumptions that are made in the methods. E.g., how important is the annual timestep, and would the results look very different when resolving the annual dynamics? How would varying input u or varying transfer matrix B (as a function of meteorology) change the conclusions? How would a representation of changes in forest structure or regeneration affect the conclusions?

The results indicate that the slow non-structural carbon pool has lifetimes that exceed those of the slowest other pools in the models (notably the wood). I would expect the authors to comment to this, as it is at least counterintuitive to have the non-structural carbon of higher age than the structural carbon. This may also be because of the underdetermination in the setup (see my comments above). Could the authors comment on that?

The figures in the appendix need to be used in the paper. Is there a difference between Fig. 6 and Fig. A5?

Minor remarks

- The abstract describes the model in very general terms before explaining the application to Harvard forest. Whereas it is fine to present the model in general terms first, I think it could help to mention early on that you are talking about forest ecosystems. You may consider adding "forest" to the title as well.

- p 3/l 12: "random paths" - I do not think that "random" is appropriate in this context; the C can follow multiple paths, but it does not move randomly through a tree. Please consider replacing.

- p 6/l 14: There is some repetition in this sentence, and the crucial information on which data sets are used appears to be missing. Please check.

- p 6/l 21: Please explain what the constraint is that is determined from "the NSC calculations from Wood". Do you have a number for the 2nd NSC pool that can be used?

- p 7/l 6: What is the parameter set with the highest frequency?

- p 9/l 8: replace "tree" with "three" (I suppose)

- p 9/l 11: replace "stronger" with "larger"

- Figure 3: One of the foliage lines appears to be invisible (or hidden behind one of the other lines). Please check.

- It seems that Fig. 4 is not referred to in the text, whereas it is crucial for the paper. Please add a reference to it.

- Figure 4: What causes the large spike in the root C stocks? Is this an initialization problem?

- Figure 7: The median for storage 1 appears to be missing from panel A.

- p 17/l 13: How do you determine the 20-year longer growth? The curves should follow an asymptotic behaviour towards the steady state.

- Figure A4: Some lines appear to be missing.

---

## Referee Comment (RC4) · Anonymous Referee #4 · 6 Oct 2017

The article develops an interesting approach to distribute structural and labile carbon into age distributions with the resulting transit time of carbon in the vegetation. They tested three different allocation models and used measured carbon values from Harvard Forest to parameterize each model and atmospheric isotopic carbon to compare against the isotopic signatures calculated from each structural component. Age and transit time distributions were different for the different allocation models, showing older age distribution in the model with more storage compartments, as well as with the wood components.

[Figure]

1. At the end of the introduction it is stated that the models are diagnosed according to four metrics, but only metrics 1 and 2 (carbon fluxes and bomb spike) are based on observations, and even those are not necessarily for each component, so the diagnosis is really just an analysis of how the different model results compare or what they imply rather than an actual metric based on observed values. There needs to be better distinction between what is actually observed vs what is modeled throughout the paper.

2. So, one justification for this research is that age and transit times can be measured with isotopic tracers, but that was not done in this study. There should be some examples in the literature of these types of measurements or some attempt to use them to verify the results of this study. It would be nice if there were some way to prove that adding more compartments improves the age distribution and transit times. However, compared to the total atmospheric isotopic signatures, the multi-compartment schemes actually do worse, which the authors attribute to perhaps the lack of phenology.

3. The models used here are purely parameterized models without any processes at all, so how would this approach actually apply to models that were process-based? Bayesian optimization of parameters has been done before, but it is impressive that this approach was taken here. Any speculation about how this age and transit distribution approach could be applied and tested in some of the major ecosystem models?

4. Non structural carbon would seem to be the same thing as labile carbon, so maybe indicate that first time the term is used.

5. The allocation models here really refer to models with different quality storage pools. The allocation itself seems to be simply constant rates – but I would consider allocation model to imply different methods of determining allocation, i.e. literature-based rates, cost-benefit analysis, etc. I would prefer more clarity on how the constant rates differ between the pools, and how they are determined.

6. P. 7, and of first paragraph: How do the functions used to calculate age and transit time distributions relate to the formulas in the introduction? Last Paragraph: What is

meant by lower-diagonal with respect to the figure?

7. P. 9, Figure 3: Are the observations data the dots and vertical lines? Seems like a large range for the error bars – why so large, and what are they based on? Table 2: What is the criteria for positive or negative correlations (i.e. how large and R2 value?)

8. Figure 4 is not mentioned in the text. More detail on the source of the isotopic carbon data would be useful. How are the "bomb spikes" used to determine radiocarbon values?

9. P. 11: Not sure whatis meant by "Notice that distributions with the highest peak (occurred) closer to 0 years, and with younger mean and median ages had the youngest C".

10. P. 16, second paragraph: What parameters are the ones that compose the beta vector and B matrix? Not all the appendix figures (A4 to A9) are mentioned in the text. At bottom of page, what are implications of the different age values – i.e. how do the 1.71/2.14 and 45 yr relate to the 7.2?

---

## Author Comment (AC1) · 2 Nov 2017

We appreciate the time that Referee 1 dedicated to review our manuscript. In the text below we quote the referee's comments in italics and provide our response below in blue:

*General comments : The research article 'Ages and transit times as important diagnostics of model performance for predicting carbon dynamics in terrestrial vegetation models' promotes using the distribution of C ages and C transit times of different tree*

[Figure]

*organs for improving performance of vegetation model. In order to do so, the authors tested three different carbon allocation schemes into a simple vegetation model with the aim to discriminate the three allocation models in terms of C stock, C flux, radio-carbon, C ages and C transit times distribution. The paper concludes that C ages and C transit times distribution can indeed be used to evaluate the skill of the allocation schemes. Furthermore, the authors encourage the scientific community to use their approach for future model comparison or validation. In my opinion the authors did a good job in developing a powerful method to help reducing uncertainty in model output as well as facilitating model development.*

We appreciate the motivating comments from Referee No. 1. This referee's impression from the manuscript captures what we intended to communicate with it.

*I really appreciate that the R scripts are clear and easy to use by someone interested in applying this method to their own model. I'm convinced that the ideas presented in this paper our of interest to the readership of biogeosciences, but the presentation itself needs to be improved. The methods and result sections lack essential information. Before resubmitting the manuscript, a senior researcher should carefully edit these sections such that they meet the minimal requirements for publication.*

We agree that there were some explanations missing and that there were certain points of the methods section that could be improved. Thus, we made major modifications to this section, which can be found in the suplementary material of this response.

*Furthermore, the following concerns should be addressed in the manuscript:*
*1) The study relies on theoretical simulations from an unrealistic vegetation model to match the requirements of the mathematical calculation of ages and transit time. I un-derstand and agree with the need for this approach. This caveat should, however, be addressed in the discussion and the conclusion where it was argued that the method can be useful for evaluating more complex vegetation models. In my opinion this sug-gestion is overly simple and this statement is not needed for the study. I would rec-*

*ommend the authors to stick to the more theoretical conclusions. Based on my under-standing of complex vegetation models, the numerical and conceptual work required to implement the proposed method into a vegetation model with full complexity is too large to be downplayed. If the authors disagree, I would expect at least a paragraph dedicated to how this functionality could be integrated into a model with phenology, environmental changes, forest management i.e. forest structure and genetics. This would really enhance the impact of the manuscript as it may convince modellers to apply your method in their models.*

The explanation on how this approach could be applied to more complex models was shortly examined on page 17 line 17. In order to calculate mean ages and transit times of non-autonomous models (inputs and process rates change over time), we would need a different set of equations. Rassmusen et al. (2016) have provided equations for the mean age and mean transit time for non-autonomous models, which require knowledge on the entire history of inputs and cycling rates for the duration of the simulation. This information is not available for Harvard forest, and for this reason we did not include these computations in our analysis. However, for complex models in which process rates vary over time due to phenology, environmental change, forest management, etc., one would obtain nevertheless age and transit time distributions. These distributions would be time-dependent, i.e. one would obtain a different distribution for each time-step. Since this manuscript is an introduction to the main concept of age and transit time distributions in vegetation models, we believe it is better to keep it simple and show these distributions for the steady-state case only. This would allow the reader to grasp the main concept, and if needed, apply to the more general case in which process rates change over time.

*2) The method section is concise, which in general I like, but the current description of the methods is too concise making it hard to follow. Each time the authors refer to a method described by another study they referring to it without a explanation or at least the reason why they choose this one. As a general guideline readers should be able*

*to understand the method section without having to consult any other papers. What justifies the use of a two-step instead of one step optimization? Why do you need Bayesian optimization?*

As mentioned before, we admit that the Methods section needed some improvements and we have worked on that for the new version of the manuscript. The Bayesian optimization gave us the possibility of exploring the parameter space, because it searches for alternative parameter sets that can result in a good fit of the model to the data. In order to have a good starting point for this search, we used the output of the classical optimization as an initial constraint. The parameter sets that we obtained were used in the uncertainty analysis, as explained in the P7L10.

*Why do you test co-linearity between parameters (answered too late in the discussion)*

We agree that the explanation of why we test collinearity should be included in the methods, specially because it is not a common practice in the field. The new version of the manuscript includes a rationale for the use of the collinearity analysis early in the Methods section.

*3) Since the authors use a lot of R packages to manage complex statistical methods, I guess that they are familiar with statistics and coding. Surprisingly, simple tests such as the t-test, anova, or metrisc like Root Mean Squared Error are not used to highlight their finding. The conclusions are not backed-up by any statistics, which is mandatory for publication in a journal like Biogeosciences.*

This is an interesting point because it is customary to support empirical findings with statistics. However, we are dealing with a completely different case here. In empirical studies one has data that is assumed to be random draws from an unknown distribution. Here, we have known distributions and no samples are drawn from them, only calculated mean values from the distributions. Unfortunately, in this case the usual t-test or even the Kolmogorov-Smirnov test for comparisons of samples from different distributions are not appropriate because we are not dealing with samples whose distribution is unknown; we are actually within the rare case in which the distribution is known and we not only have it's equation, but also the parameter values. We have to admit that we are not aware of any statistical test designed to compare two or more distributions without having to draw samples from them. But even if there would be a test, the results would be trivial. We know that the distributions are different because we have different parameter values from all of them. Therefore, we would always reject the null hypothesis that the distributions are different. Since this would be a trivial result, we refrain from adding any statistical comparison to our analysis.

*Detailed comments on the tables and figures:*
*Table 1: the column called 'final' is not clear! Why this column differs from 'Best2' column ? I guess it is because of the constraint optimization on C stock but it is not described in the caption. Do not show the median and the quantiles of Best2 if you will not use these values in the next step.*

We agree that the names that we used for the parameter sets were confusing. We now describe them in a better way in table 1 as well as in the Results section. 'Final' is the parameter set that was most frequently chosen by the Bayesian optimization method and was used for all of the simulations, unless otherwise noted. Furthermore, the above mentioned 'quantiles' are the quantiles of the distribution of the values of each parameter, as a result of the parameter space exploration performed with the Bayesian optimization.

*Table 2:I am not sure this one is necessary ...*

We moved it to the appendix

*Figure 3: Two figures instead of one can be better to avoid the compressing of the leaf C stock.*

This is a good point, we followed the recommendation of the reviewer here.

*Figure 5,7: Put some statistics to highlight your results. You argue that mean ages and*
*transit time are different across allocation model. Support it with a probability! Figure 8: Again show that mean ages are significantly different.*

Please see our response above regarding the applicability of statistical tests in this case.

We hope that we addressed all comments from Referee 1 adequately, and improved the clarity of this manuscript.

Please also note the supplement to this comment:
https://www.biogeosciences-discuss.net/bg-2017-308/bg-2017-308-AC1-supplement.pdf

---

## Author Comment (AC2) · 2 Nov 2017

We appreciate the time that Referee 2 dedicated to review our manuscript. In the text below we quote the referee's comments n italics and provide our response below in blue:

*Despite an encouraging title and a promising abstract, I find the study led by Ceballos-Núñez & al. quite disappointing and I doubt that the modelling approach proposed here can be used by a larger community to constrain carbon dynamics in terrestrial*

[Figure]

*vegetation model.*

It is unfortunate that the reviewer was disappointed with this manuscript, but we are convinced that this issue can be easily resolved, because there seems to be certain misunderstandings that will be unveiled in the following points.

*The "vegetation" model described in this study is in fact a very simple box model, calculating fluxes between different carbon storage compartments, as well as the carbon stock of each compartment, comparing three model structures (i.e. increasing the number of carbon pools). The model was forced by a constant input of carbon (GPP=1400 gC m-2 year-1) and run on a yearly time step, with no change in environmental forcing (climate, CO2,etc.). The results are shown during the transient spin-up (e.g. fig4) or at steady state (most figures). In my opinion, this approach (i.e. yearly time step, constant GPP, no external forcing) is absolutely not appropriate for "predicting carbon dynamic" as claimed in the title. The actual dynamic of the carbon cycle, i.e. the increasing terrestrial carbon sink, is happening because of transient changes in external environmental conditions affecting the terrestrial carbon dynamics. Therefore it is impossible to draw any conclusion from this study, with respect to the actual dynamic of the system.*

It is important to clarify that autonomous systems, as those modeled here, are still dynamic, and an useful tool to assess processes that occur within the vegetation, which in this case is the distribution of carbon among different compartments. If we were interested in predicting the effect of a specific disturbance such as time-varying atmospheric CO2 or temperatures, we could still predict time-varying ages and transit times distributions. However, this is not the objective of our study; we are rather interested in presenting **the concept** of ages and transit times as useful diagnostics of model performance and as a tool to explain mixed ages of non-structural carbohydrates previously reported in field studies. For the case of transient simulations, which the reviewer advocates here, formulas do exist to calculate mean ages and transit-times (see Rasmussen et al. 2016), and if we would have knowledge on the time evolution of process

rates at the Harvard Forest for the simulation period, we could have calculated the time evolution of age and transit time distributions. But as an introductory paper on the main concept, we do not consider appropriate to include the additional complexity inherit of the time-evolving formulas. For this reason, we decided to use the autonomous case to introduce our concept.

*I also disagree on the way the authors claim they evaluated their model using obser-vation. [i.e. 'We found a good fit of the three model structures to the available data' [abstract]]. First, I understand that the data used for model "evaluation" are the same that the one used for model optimisation as these are the only data mentioned in the manuscript. Is the model calibrated against Harvard Forest data (i.e. results in Table 1) and then compared to the same data for evaluation (Figures 3 and 4)? Or did I miss something? Second, there is no information on how the simulations were done for evaluation.*

This is a good point with regard to the "evaluation of the models", since in case that we actually wanted to evaluate them we should have used another data set. However, we actually never mentioned that we evaluated the models. In the figure 3 we simply showed that the model simulations fitted the data points, but this is only to give an idea that the predictions of C stocks are in accordance to a particular forest. We under-stand that this might be a source of confusion, but it is important to highlight that this work is a theoretical exercise, and the fit of the models to the data is only to have a rough example that can be related to a 'real' forest. We made our intentions clearer in the methods and in the results, as can be seen in the material that we included as supplement of this response.

*Figure 4 clearly shows that the model is in transient conditions from 1950 to 2010, with wood carbon stocks increasing and being comparable to the observations in 2010. That would make some sense if the model actually started in 1950, with external forcing (climate, CO2, land use, etc.) changing from year to year. My understanding is that this is not the case here. The model is simply spinning up, slowly reaching steady state.*

*The agreement in 2010 is hence completely artificial. Cstock wood does not seem the have reached equilibrium, running another 100 years and it would be well above the observations. Unless I missed these two elements, there is strictly no evaluation in this paper.*

Again, we are not interested here in finding the best model that reproduces the entire history of C accumulation for the Harvard Forest site as modified by changes in atmospheric CO2 concentrations and climate change, but rather to find a set of realistic parameters that at least can reproduce the trend in carbon accumulation for some of the measured pools in this site. The data shows that this forest is not in equilibrium yet, and our transient simulation approaches these dynamics well. We consider this is enough to obtain a set of parameters that allows us to show some examples of the main concepts we want to introduce: age and transit time distributions of carbon. Please keep in mind that this is a conceptual paper, and we make no claims regarding the accuracy of the predictions for the specific site. We are rather interested in introducing a new set of model diagnostics that can be very useful for more specific simulations.

*Finally the co-authors conclude that 'Differences in model structures had a small impact on predicting C stocks in ecosystem compartments, but overall they resulted in very different predictions of age and transit time distributions'. I will argue here that considering the fact that each of their model parameters was constrained using the same carbon stocks, this is a result one should expect as a direct outcome of their methodology and no conclusion on the reality of the processes can be drawn from it.*

We agree that sentences such as the one cited can be interpreted literally as "the model structures had a small impact on predicting C stocks differences". However, the C stocks were not listed as one of the metrics that we used. Thus, what we meant was that although all the models had similar predictions in C stocks, they had important differences with regards to other metrics. To avoid this confusion, we made our point clearer in the abstract, results and discussion. It is anyway noteworthy that the results

of the sensitivity analysis show that these data is an insufficient constraint, since different combinations can result in the prediction of similar C stocks, hence the equifinality section.

We hope that we addressed the comments of Referee 2 adequately and with that improved the clarity of this manuscript.

Please also note the supplement to this comment:
https://www.biogeosciences-discuss.net/bg-2017-308/bg-2017-308-AC2-supplement.pdf

**Supplement:**

[revised manuscript text omitted]

---

## Author Comment (AC3) · 2 Nov 2017

We appreciate the time that Referee 3 dedicated to review our manuscript. In the text below we quote the referee's comments in italics and provide our response below in blue:

*The manuscript by Ceballos-Núñez et al. describes a study on transit times of C through a temperate forest, with the help of a framework with a number of pools and transfer time coefficients between these pools. The number of non-structural pools is*

[Figure]

*varied to investigate the importance of these for representing the transfer. Whereas the concept used in the manuscript is generally well explained, more information should be provided on the details. I have some reservations to the chosen methods, but this may also be partly caused by misunderstanding due to the too concise description. Some parts of the study need to be clarified, and slight alterations to the methodology may be required. This manuscript can become an interesting contribution to the literature on modelling carbon allocation in (temperate forest) ecosystems, but I would recommend the authors to clarify the methodology, and also to rethink some of the concepts that they apply in their setup. I hope that the comments below can support the authors in revising their work.*

We thank referee 3 for an accurate summary of our manuscript. We agree that in the concise Methods section we left out some important details that could have increased the clarity of our work. Therefore, we made major modifications to the Methods section, which we have included as a supplement to this response.

*Major remarks*
*A general issue with the methodology is the limited amount of observations that is used to constrain the transfer matrix B. Firstly, these data should be described more carefully, e.g., Figures 3 and 4 seem to suggest that wood or biomass numbers for multiple years are available, which does not appear in the Methods section. Secondly, and more importantly, it seems to me that the available data make it impossible to separate the non-structural pool(s) in the models, which leaves the system underdetermined. The authors also seem to refer to this in their methods (p 7/l 5). It is unclear how a Monte-Carlo setup for sampling the parameter space could help to constrain these, and in fact the authors seem to suggest that it does not. I would suggest to either explain more carefully how the Monte-Carlo setup was applied, which parameters were estimated with it, and which constraints were used, or to skip the Monte-Carlo setup and replace it by a more simple parameter estimation based on literature values, which seems more appropriate for a system that cannot be constrained by observations.*

We agree with the reviewer in that there was limited data to constrain values for the transfer matrix **B**, and that we did not carefully explained what data was used for the parameter estimation procedure. We included a description of the data that we used and the way in which we calculated the C stocks from it, which addresses the concern on the use of the wood biomass numbers. However, the issue of limited availability of observations is an issue that remains and for which we can do little about. We did in fact used reported values from the literature to fix some of the parameter values (P6L11) or to set ranges for the parameter optimization routines. Nevertheless, we had non-identifiable issues and we think it is important to report them in our analysis in a transparent way. In this sense, the Bayesian optimization we performed, helped us to honestly report uncertainty ranges for possible parameter values, but it is in no way a method to fix the non-identifiable problems we found. We think that by reporting uncertainty ranges for the parameters and the predictions we can better deal with the uncertainty related to the lack of observations and values from the literature.

*Although the setup of the three steady-state models is generally well described in the methods, there is one aspect that I have not been able to resolve with regard to the photoassimilates pool. The text seems to suggest that this is not a real pool but just partitions the input vector u to the other pools and to autotrophic respiration, leaving it in essence a flux (or sum of fluxes) rather than a pool. However, in Fig. 2, $C_P$ receives its own box, and the presented results (e.g. Fig. 8) indicate that the photoassimilates pool also has an age. If this is indeed the case in the model, there would be one nonstructural pool (namely, $C_P$) in Storage 0, and two respectively three non-structural pools in Storage 1 and Storage 2. Maybe the authors can clarify whether this is the case or not. If it is, I would recommend to remove it as a pool, to reduce the size of the vector x and thereby reduce the number of parameters that need to be estimated.*

This is a compartment that we decided to keep from the original model proposed by Richardson et. at. (2013). This is in fact not a storage compartment, it is one of the two types of foliage compartments. Since the foliage is divided into photoassimilates and

structural foliage, the former is strictly a compartment, and has its own state variable. However, this compartment does retain the carbon for a short time, giving the impression to be a component that partitions the input flux. Since this confusion might have been originated from the model schemes in figure 2, we changed the name of the variable $C_f$ to $C_{Strf}$ and we included a short statement about these two compartments in the legend of that figure. Also, to clarify, $u$ is not a vector, it is a scalar, the partitioning vector is $\beta$.

*The discussion section describes the limitations of the method to some extent, but it should also be used to discuss the validity of the assumptions that are made in the methods. E.g., how important is the annual timestep, and would the results look very different when resolving the annual dynamics? How would varying input u or varying transfer matrix B (as a function of meteorology) change the conclusions? How would a representation of changes in forest structure or regeneration affect the conclusions?*

Thanks for the suggestion. Indeed, resolving annual dynamics and making the system time-depend would not have any impact on the main conclusions we derived from this study: 1) age and transit time distributions strongly depend on different carbon allocation schemes imposed by the model structure, and 2) observed mixes of carbon age in non-structural carbohydrate pools can be easily explained by the existence of age distributions in vegetation compartments. Explicitly resolving intra and inter-annual dynamics simply result in time-varying age and transit time distributions, but does not invalidate the existence of these distributions. We added a few sentences to the discussion to address this comment.

*The results indicate that the slow non-structural carbon pool has lifetimes that exceed those of the slowest other pools in the models (notably the wood). I would expect the authors to comment to this, as it is at least counterintuitive to have the non-structural carbon of higher age than the structural carbon. This may also be because of the underdetermination in the setup (see my comments above). Could the authors comment on that?*

We appreciate this detailed remark. Actually this is only an artifact of cropping the ranges of the distributions in the plots. Since this is clearly causing a confusion, we extended the ranges, so that it is possible to see that the tail of the age distribution for the wood compartment is longer.

*The figures in the appendix need to be used in the paper. Is there a difference between Fig. 6 and Fig. A5?*

Thank you for this remark; we added the missing references. The difference between figures 6 and A5 is the parameter set with which we run the models to obtain them; for the simulations presented in figure 6, each model was run with a different parameter set, while for figure A5 the same parameter set was used for the three models.

*Minor remarks*
*- The abstract describes the model in very general terms before explaining the application to Harvard forest. Whereas it is fine to present the model in general terms first, I think it could help to mention early on that you are talking about forest ecosystems. You may consider adding "forest" to the title as well.*

Although the examples presented in our manuscript correspond to a forest, the approach that we are suggesting is applicable to other systems. The use of age and transit time distributions as diagnostics can be implemented to models portraying other systems, as long as they follow the assumptions explained in the formulas. For this reason we kept the description of the model general enough so it can be applied to other systems.

*- p 3/l 12: "random paths" - I do not think that "random" is appropriate in this context; the C can follow multiple paths, but it does not move randomly through a tree. Please consider replacing.*

We understand that the term 'random paths' can be misinterpreted in the context of tree physiology. This is why we edited that line to read as follows: At each time step,

a particle may randomly stay where it is, or flow to the next compartment with a rate (speed) given by the transfer coefficients (also know as cycling rates).

*- p 6/l 14: There is some repetition in this sentence, and the crucial information on which data sets are used appears to be missing. Please check.*

We are grateful for the identification of this issue. We indeed had the same title for different links. We reduced the redundancy and explained more what kind of data we used.

*- p 6/l 21: Please explain what the constraint is that is determined from "the NSC calculations from Wood". Do you have a number for the 2nd NSC pool that can be used?*

We explained better that we calculated the C stocks of NSC by using equations that derive it from the C stocks in wood.

*- p 7/l 6: What is the parameter set with the highest frequency?*

The parameter with the highest frequency is the one selected more often by the Bayesian optimization method during the parameter exploration. We clarified this in the text.

*- p 9/l 8: replace "tree" with "three" (I suppose) - p 9/l 11: replace "stronger" with "larger"*

We appreciate this remark, we made the corrections.

*- Figure 3: One of the foliage lines appears to be invisible (or hidden behind one of the other lines). Please check.*

There was an overlap between the lines, so we fixed it by dividing that plot into two separate ones, one for wood and the other for foliage.

*- It seems that Fig. 4 is not referred to in the text, whereas it is crucial for the paper. Please add a reference to it. - Figure 4: What causes the large spike in the root C*

*stocks? Is this an initialization problem?*

In fact we referenced the figure, but only with a number, which is probably the reason why it was overlooked. We have corrected that. Also, it is probably true that the cause of the large spike in the root C stocks might be due to having the wrong initial values for that compartment.

*- Figure 7: The median for storage 1 appears to be missing from panel A.*

The line of the median was hidden behind the mean line of another model, we corrected this by lowering the range of the x-axis.

*- p 17/l 13: How do you determine the 20-year longer growth? The curves should follow an asymptotic behaviour towards the steady state.*

The age and transit time distributions were obtained under the assumption that the models were in steady state. Thus, we can say that the mean ages are predictions of the mean ages of the vegetation once it reaches steady state. Since in each model the mean ages are different, we can consider that their difference is the remaining time that it takes them to reach steady state.

*- Figure A4: Some lines appear to be missing.*

In this case it is more complicated to fix the overlap because an outcome of running the three models with the same parameter set was that they all had similar mean and medians in their distributions.

We hope that we addressed the comments of Referee 3 adequately and with that improved the clarity of this manuscript.

Please also note the supplement to this comment: https://www.biogeosciences-discuss.net/bg-2017-308/bg-2017-308-AC3-supplement.pdf

---

## Author Comment (AC4) · 3 Nov 2017

We appreciate the time that Referee 4 dedicated to review our manuscript. In the text below we quote the referee's comments in italics and provide our response below in blue:

*The article develops an interesting approach to distribute structural and labile carbon into age distributions with the resulting transit time of carbon in the vegetation.They tested three different allocation models and used measured carbon values from Har-*

[Figure]

*vard Forest to parameterize each model and atmospheric isotopic carbon to compare against the isotopic signatures calculated from each structural component.*

We are glad that the referee found the article interesting, but we have to clarify that we did not 'distributed the carbon into age distributions'; we calculated the age and transit time distribution of carbon particles in a vegetation system and its compartments.

*Age and transit time distributions were different for the different allocation models, showing older age distribution in the model with more storage compartments, as well as with the wood components.*
*1. At the end of the introduction it is stated that the models are diagnosed according to four metrics, but only metrics 1 and 2 (carbon fluxes and bomb spike) are based on observations, and even those are not necessarily for each component, so the diagnosis is really just an analysis of how the different model results compare or what they imply rather than an actual metric based on observed values. There needs to be better distinction between what is actually observed vs what is modeled throughout the paper.*

We interpret this and the previous comment as a sign that we need to make our Methods section clearer. Please find enclosed the new version. The only measurements that we used for parameter optimization were published aboveground biomass, LMA and LAI. These measurements were only used in the optimization procedures, to find suitable parameter values. The model comparison was mainly qualitative, using simulations, e.g. the method used for the radiocarbon is clearly stated in the first line of the section 3.2.2. In the rest we discuss that we were comparing the model simulations with respect to their expected dynamics, depending whether they were fast or slow cycling components.

*2. So, one justification for this research is that age and transit times can be measured with isotopic tracers, but that was not done in this study. There should be some examples in the literature of these types of measurements or some attempt to use them to*

*verify the results of this study. It would be nice if there were some way to prove that adding more compartments improves the age distribution and transit times. However, compared to the total atmospheric isotopic signatures, the multi-compartment schemes actually do worse, which the authors attribute to perhaps the lack of phenology.*

In the Discussion we compared published mean ages of NSC with the ones predicted with the models. We agree that it would be more interesting to have measured C age distributions to compare with the simulations, but they are unfortunately unattainable with the current empirical methods. Also, one important clarification is that radiocarbon values for the different pools cannot be compared directly to atmospheric radiocarbon values as this comment seem to imply. They can only be compared to radiocarbon values measured on the tissue of the different pools.

*3. The models used here are purely parameterized models without any processes at all, so how would this approach actually apply to models that were process-based? Bayesian optimization of parameters has been done before, but it is impressive that this approach was taken here. Any speculation about how this age and transit distribution approach could be applied and tested in some of the major ecosystem models?*

The models we use are simple, which helps us to introduce the concepts of age and transit time of carbon in vegetation compartments. Independent on the complexity of the model being used, and the time steps of specific simulations, we can always expect mixes of C ages of different compartments, and this results in age and transit time distributions. The concept is independent on the complexity of the model, but the specific shape of the distributions would in fact depend on the specificity of the model. So, there is no consequence on whether the concept can or cannot be applied for more complex models, it only would affect the particular shape of the distributions.

*4. Non structural carbon would seem to be the same thing as labile carbon, so maybe indicate that first time the term is used.*

Thank you for this suggestion. We indicated that in the introduction.

[Figure]

*5. The allocation models here really refer to models with different quality storage pools. The allocation itself seems to be simply constant rates – but I would consider allocation model to imply different methods of determining allocation, i.e. literature-based rates, cost-benefit analysis, etc. I would prefer more clarity on how the constant rates differ between the pools, and how they are determined.*

We agree that there are other methods to estimate the allocation rates, and that we did not explained the reasons why we chose to estimate the parameters of our models using the Bayesian optimization method. In the new version of the Methods section we mention that we chose this method because it gave us the possibility of exploring the parameter space and with that find parameter sets to do an uncertainty analysis of the model. The goal of the parameter estimation procedure was to obtain parameter sets for each model, which could allow them to simulate similar C stocks to those obtained from the Harvard Forest Archives. The upper and lower boundaries used to constrain the parameter values during the estimation were obtained from the literature, for a similar forest. As we explained in the point 3, we decided to use simple models (with fixed coefficients) to illustrate the concept of 'age and transit time distributions'; this is why the environmental forcing was not part of the estimation of the rates. However, it is important to clarify that even if we would have used other methods to obtain the rates, we would have still obtained different ages and transit time distributions for each model'.

*6. P. 7, and of first paragraph: How do the functions used to calculate age and transit time distributions relate to the formulas in the introduction? Last Paragraph: What is meant by lower-diagonal with respect to the figure?*

The formulas mentioned in the introduction were implemented as functions in the 'R' package 'SoilR', in order to calculate age and transit time distributions. With "lower-diagonal" we were referring to the values under the diagonal. We corrected that sentence.

[Figure]

*7. P. 9, Figure 3: Are the observations data the dots and vertical lines? Seems like a large range for the error bars – why so large, and what are they based on?*

We agree that we did not explained properly the way in which we used the data obtained from the archives of Harvard Forest. We added a short explanation on how we used the measurements of aboveground biomass, LAI and LMA to calculate the C stocks that we show in figure 3. The large deviations from the C in wood is probably due to the large variation in the aboveground biomass measurements.

*Table 2: What is the criteria for positive or negative correlations (i.e. how large and R2 value?)*

The assignment of positive or negative correlations was performed based on the sign of the values observed in the pairwise plots. Only $R^2$ values $< -0.1$ and $> 0.1$ were assumed to account for correlations.

*8. Figure 4 is not mentioned in the text. More detail on the source of the isotopic carbon data would be useful. How are the "bomb spikes" used to determine radiocarbon values?*

We had actually referenced the figure, but only with a number, which is probably the reason why it was overlooked. We corrected that mistake. The calculation of radiocarbon in the different compartments relies on a standard atmospheric radiocarbon curve (Hua et al. 2013), which is used to introduce radiocarbon in all vegetation compartments at rates specified by the vector $u$ and the transfer matrix $B$. This functionality is part of the 'SoilR' packaged used here. We added a short description in the methods about the incorporation of radiocarbon in the simulations and provided a citation for additional details of its implementation in the SoilR package.

*9. P. 11: Not sure whatis meant by "Notice that distributions with the highest peak (occurred) closer to 0 years, and with younger mean and median ages had the youngest C".*

That sentence was meant to explain how to interpret the shape of the distributions and, based on that, show that the model *Storage: 0* had younger C than *Storage: 1* and *2*. However this explanation might not be needed, so we removed that sentence.

*10. P. 16, second paragraph: What parameters are the ones that compose the beta vector and B matrix?*

The elements that compose the $\beta$ vector and the **B** matrix are described in the equation (1). The components of the matrix **B** are represented by the symbols next to the arrows in figure 2, which are the parameters that control the C transfer through the compartments.

*Not all the appendix figures (A4 to A9) are mentioned in the text.*

We appreciate this remark, we have added the missing references.

*At bottom of page, what are implications of the different age values – i.e. how do the 1.71/2.14 and 45 yr relate to the 7.2?*

This is the part that we mentioned in the response to the comment 2. We were comparing the simulated mean ages of the NSC to the ones obtained from literature. The published values had a wide range. Thus, although the mean ages of the storage compartments with fast cycling were lower than the mean published value, they were still within the range.

We hope that we addressed the comments of Referee 4 adequately and with that improved the clarity of this manuscript.

Please also note the supplement to this comment:
https://www.biogeosciences-discuss.net/bg-2017-308/bg-2017-308-AC4-
supplement.pdf
* * *

---

## Author Response (AR2)

**Max–Planck–Institut für Biogeochemie**
**Max Planck Institute for Biogeochemistry**

MPI für Biogeochemie · Postfach 10 01 64 · 07745 Jena, Germany

Max-Planck-Institut
für Biogeochemie

Editors
Biogeosciences

**Verónika Ceballos-Núñez**
Tel.: +49-(0)3641-57-6143
vceball@bgc-jena.mpg.de

1st February 2018

Dear Dr. Akihiko Ito

Thank you for your comments, and for giving us another opportunity to revise and improve our manuscript. Thanks also for recognizing that the previous version was fairly improved despite the lack of statistical comparisons, which for this particular case are not appropriate given that we actually know the complete distributions of ages and transit times. We appreciate that you recognize this, and that the manuscript is a valuable conceptual study.

This new version of the manuscript addresses the spike in root carbon that reviewer 3 and you pointed out. We made the suggested corrections of the initial values of root C, and consequently obtained new simulations. This means that all of the original plots and tables have slightly changed, and therefore we are submitting a revised version that includes new versions of most figures. These changes, however, had no impact on the overall message of the manuscript.

We hope that we addressed this new version is suitable for publication.

Sincerely,

Verónika Ceballos-Núñez

Max-Planck-Institut für
Biogeochemie
Hans-Knöll-Straße 10
07745 Jena
Germany

Tel.: +49-(0) 3641 / 57−6110
Fax.: +49-(0) 3641 / 57−6110
http:// www.bgc-jena.mpg.de

Direktorium
Markus Reichstein (Managing Dir.)
Martin Heimann
Susan Trumbore
ID-Nr. DE 129517720